# A Controlled Synthetic Benchmark for Educational Aspect-Based Sentiment Analysis

## Abstract

Large language models are increasingly used to synthesize labeled training data where annotation is scarce, but the generated labels are often unfaithful to the text, and it is difficult to tell whether a synthetic benchmark carries genuine learnable signal or merely reproduces label frequencies. We study both problems in educational aspect-based sentiment analysis (ABSA), a setting where real aspect-labeled feedback is private and costly to annotate, with a methodology that applies beyond it. We release a controlled corpus of 10,000 synthetic course reviews over a 20-aspect pedagogical schema, generated by sampling supervision targets separately from nuance attributes, so the same labels recur across varied course contexts and styles. We then introduce a faithfulness-aware pipeline: a cost-matched LLM audit scores how well each declared label is supported by the text, and that score both filters the training data and is itself validated across independent LLM judges and against human labels (Cohen's kappa 0.56 on two annotated corpora). Three controls establish that the corpus carries learnable signal rather than label priors: permuting the labels collapses detection to the trivial floor (0.182 versus 0.276 micro-F1), accuracy scales monotonically with training size (0.183 to 0.285), and restricting to faithfully labeled rows raises the ceiling. Evaluated by transfer to real human-annotated feedback (the unbiased metric), faithfulness-aware row filtering lowers transferred sentiment error on Herath across two architectures (paired 95

## 1 Introduction

Consider a typical undergraduate program at the end of a teaching term. Every course generates dozens to hundreds of free-text comments alongside numerical ratings, and the program lead reads, in practice, a tiny fraction. A 4.1 / 5 satisfaction score does not say whether students appreciated lecture clarity or merely tolerated it, whether exam fairness held up while workload climbed, or whether the same TA team was praised for support and criticised for grading transparency. Aspect-based sentiment analysis (ABSA) is the layer that translates raw comments into the per-dimension, per-instructor, per-cohort signals a program lead can act on. An institutional ABSA pipeline of the kind this paper enables would let a department, in one term-over-term comparison, see that `tooling_usability` sentiment across an introductory programming sequence dropped sharply after a submission-platform change, or that `pacing` declined in a senior elective as the syllabus accumulated topics without trimming.

Three institutional workflows benefit directly from such a pipeline. **End-of-term review**: an aspect-tagged feed converts an unread comment pile into per-aspect dashboards a program lead can scan in minutes, with negative-rate signals routed to the right intervention (TA training for `support`, syllabus rewrite for `pacing`, platform fix for `tooling_usability`). **Mid-term pulse**: a short free-text survey at week six, tagged by aspect, surfaces the categories with rising negative density (often `pacing` or `support`) early enough that the instructor can still intervene before final assessment. **Longitudinal monitoring**: the same course taught over four to eight semesters yields an aspect-time series whose drift exposes slow-moving problems (workload creep, declining `peer_interaction` as cohorts grew, gradually narrowing `practical_application`) that no single end-of-term reading reveals. None of these workflows demand a perfect classifier; they demand one that is available, reproducible, and substantially cheaper than the human reading they replace. The bottleneck

today is not modelling architecture but supervision: educational ABSA datasets are scarce, expensive to annotate, and rarely public.

Student reviews contain actionable evidence about workload, clarity, support, fairness, materials, and overall learning experience, yet most institutional feedback analysis still depends on manual reading or coarse sentiment summaries. For pedagogy, this matters because interventions rarely target "sentiment" in the abstract; instructors and program leaders need to know whether learners are reacting to assessment design, instructional clarity, course relevance, staff support, or the overall classroom experience. Feedback research in higher education repeatedly shows that comments become educationally useful when they can be connected to actionable teaching conditions rather than treated as isolated satisfaction signals [20, 21, 22]. ABSA is therefore well matched to this setting because the same review can simultaneously praise one aspect and criticize another. The challenge is that high-quality educational ABSA datasets are scarce, expensive to annotate, and often limited to a single institution, course type, or annotation scheme.

That scarcity is not accidental. Real student-feedback data are difficult to release because they are tied to institutional processes, often contain identifying context, and require laborious annotation decisions about implicit aspects, mixed sentiment, and local pedagogical terminology. Existing educational text-mining studies show that open-ended feedback is useful for course improvement, but the underlying data are often confidential, institution-specific, or too small for straightforward public reuse [16, 23, 24]. Even when institutions collect large amounts of feedback, the resulting labels are rarely public, rarely harmonized across universities, and rarely aligned with the fine-grained aspect schemas needed for ABSA. The present study addresses that bottleneck by pairing synthetic data generation with downstream ABSA modeling. Rather than treating data generation and model training as separate tasks, it formulates them as one end-to-end system: a synthetic review generator produces multi-aspect course reviews in multiple student personas, and an analysis pipeline learns aspect detection and sentiment scoring from those labels.

The contribution is therefore both a resource and a methodology. **The resource is a controlled synthetic educational ABSA corpus whose labels are audited and faithfulness-scored; the methodology is an audit-filter-validate pipeline for LLM-generated supervision: a cost-matched audit measures per-label faithfulness, that score selects training data, and a battery of controls tests whether the resulting benchmark carries genuine signal rather than label frequencies.** Although we instantiate and evaluate the pipeline on educational ABSA, where labeled data are unusually scarce, it applies to any task trained on LLM-synthesized labels.

The remainder of the paper proceeds as follows. Section 2 situates the study in the ABSA, synthetic-data, and educational-feedback literatures. Section 3 describes the corpus design and generation protocol, Section 4 defines the benchmark task and evaluation setup, and Section 5 presents the main benchmark, external-validation, and generator-faithfulness results before Section 6 discusses limitations and implications. Generator realism-validation and corpus-scale output-control diagnostics are reported in the appendix.

## 1.1 Contributions

- **Controlled synthetic ABSA resource:** a released 10,000-review corpus with fixed train/validation/test partitions over a 20-aspect pedagogical schema, generated by separating supervision targets from nuance attributes so the same aspect labels recur under varied course contexts and writing styles rather than collapsing into one prompt pattern.
- **Faithfulness-aware data-quality methodology:** a cost-matched LLM audit that scores per-label faithfulness and acts as an actionable training-data filter, with a lowest-faithfulness negative control that collapses transfer across two architectures and two real targets, a size-matched filtering gain on the Herath target, and direct validation against human labels (Cohen's kappa 0.56 on two annotated corpora), establishing a general recipe for quality-controlling LLM-supervised data, not only this corpus.
- **Signal-validity battery:** label-permutation, trivial-floor, learning-curve, and clean-label-ceiling controls that distinguish genuine learnable signal from label frequencies, providing a reusable check that a synthetic benchmark is more than memorized priors.

- **Real-data transfer evidence:** on two human-annotated corpora (Herath, EduRABSA), synthetic-only training recovers about 60% of a real-trained model with no real labels, and synthetic pre-training followed by real fine-tuning exceeds real-only training.

## 2 Related Work

This study draws on three literatures that are usually discussed separately: general ABSA benchmarks and models, synthetic text generation for supervised NLP, and educational feedback analysis. The connection among them is central to the paper. General ABSA work provides the task formulation and evaluation vocabulary; synthetic-data research motivates controlled label generation when annotation is scarce; and educational feedback studies explain why the target problem matters pedagogically and why real labeled data are unusually difficult to assemble. SemEval task definitions helped establish ABSA as a standard fine-grained sentiment problem by formalizing aspect detection and aspect sentiment prediction with shared datasets and evaluation procedures [2, 3]. More recent ABSA work continues to show the value of pretrained transformers and sentiment-aware pretraining for fine-grained polarity modeling [4, 5].

On the data side, the synthetic-data component belongs within a broader NLP literature on augmentation, weak supervision, and LLM-based example generation. Data-augmentation surveys show that gains depend on whether generated instances preserve task semantics and enlarge useful coverage of the training distribution rather than merely adding lexical variation [25]. More recent work on LLM-generated supervision shows both promise and caution: synthetic examples can improve downstream classification when prompts, controls, and filtering are carefully designed, but task difficulty and label faithfulness remain decisive constraints [6, 26, 27]. A recent line of work formalizes this as a generation, curation, and evaluation pipeline [34] and turns directly to controlling the quality of LLM-supervised data: synthetic ABSA generators add label-refinement modules for few-shot transfer [32], LLM-generated samples help ABSA in low-resource regimes [33], and downstream classifiers can be calibrated against LLM-generated label noise after training [37]. The present work sits on the data side of that pipeline: rather than refining labels inside the generator [32] or correcting the classifier downstream [37], it audits each declared label for textual support, filters the corpus before training, and validates that audit against human labels. This caution is especially relevant here because educational ABSA is a subjective and domain-specific task whose label space often reflects local pedagogical practice rather than a universal ontology.

In education, prior work has emphasized that student feedback is noisy, stylistically varied, and pedagogically important. A broader higher-education feedback literature also shows that formative feedback matters because it shapes student learning rather than merely administrative satisfaction tracking [1]. Welch and Mihalcea demonstrated the value of targeted sentiment analysis for student comments [7], while Chathuranga et al. released an annotated student course feedback corpus for opinion targets and polarity [8]. Herath et al. later reported a 3,000-instance student feedback corpus with annotations for aspects, opinion terms, and polarities, illustrating both the feasibility and the cost of building educational sentiment resources [11]. Nikolić et al. showed that ABSA can be informative for higher education reviews, but that source characteristics matter and performance varies across aspect frequencies and review sources [9]. Misuraca et al. repositioned opinion mining as an educational analytic rather than only a text-classification exercise, arguing that open-ended comments add value beyond numeric ratings [16]. More recent reviews synthesize the educational sentiment-analysis landscape and reiterate recurring bottlenecks around annotation cost, domain language, multi-polarity, and deployment in authentic educational settings [10, 17]. A recent systematic review of aspect-based sentiment analysis in MOOC settings similarly catalogs the aspect schemas and methods used across educational platforms [38]. Together, this literature supports the relevance of the problem, clarifies why real labeled educational data are difficult to obtain at scale, and explains why a synthetic educational ABSA resource may be useful even when realism validation remains a separate concern. Table 1 positions the present study against these prior educational-feedback resources.

The dataset is intentionally more aspect-rich than narrow course-review setups that focus mainly on instructor praise or overall satisfaction. Its 20 aspects span workload, clarity, exam fairness, lecturer quality, relevance, interest, support, materials, overall experience, feedback quality, assessment design, pacing, organization, practical application, tooling usability, accessibility, grading transparency, peer interaction, and prerequisite fit.

Table 1. Educational feedback resources used to position the present study within prior work on student comments, annotated course feedback, and higher-education review analysis.

| Study | Data source and focus | Label granularity | Relation to this study |
|---|---|---|---|
| Welch and Mihalcea [7] | Student comments | Targeted sentiment | Establishes pedagogical relevance, but not a public multi-aspect benchmark |
| Chathuranga et al. [8] | Student course feedback | Opinion targets and polarity | Shows feasibility of real educational annotation, but with narrower task scope |
| Herath et al. [11] | Student feedback corpus | Aspect, opinion-term, and polarity annotations | Closest educational benchmark reference and strongest evidence of annotation cost |
| Nikolić et al. [9] | Higher-education reviews | Aspect-based review analysis | Shows source sensitivity and motivates explicit realism validation |
| This study | Synthetic course reviews with separate realism and mapped real-transfer validation sets | 20-aspect benchmark corpus, controllable prompt protocol, realism study, and conservative 9-aspect transfer check | Positions the contribution as synthetic supervision with initial external validation rather than a replacement for real annotated corpora |

This wider aspect space matters pedagogically because the actionable question for instructors is rarely whether a course is "good" in the abstract; it is which design dimension is helping or harming student learning. It also matters methodologically because a broader aspect inventory creates a harder ABSA benchmark, especially for more ambiguous categories such as relevance, interest, overall experience, support, and prerequisite fit.

The novelty claim in this setting comes from the combination of pieces rather than from synthetic text alone. The study couples *controlled* synthetic generation with an explicitly educational 20-aspect schema, a train-validation-test benchmark protocol, and external validation on mapped student feedback. In other words, the contribution lies in the combination of task framing, controllable generation, educationally meaningful aspect design, and benchmark integration.

Synthetic augmentation and few-shot prompting also provide a natural methodological backdrop for this work. Prior augmentation research in ABSA shows that semantics-preserving edits can improve downstream learning without destroying polarity structure [12], while few-shot prompting methods demonstrate that label recovery can be possible even with very limited demonstrations [13, 18]. More recent instruction-tuned and weak-supervision studies further suggest that prompt-based ABSA can become competitive when label scarcity is severe, but that careful task formulation still matters [28, 29]. This makes it reasonable to compare trainable ABSA models with zero-shot and few-shot generative baselines on the same synthetic benchmark, even if the main paper contribution remains the dataset and the training pipeline rather than prompt engineering alone.

Generative ABSA work also supports this comparison more directly. Unified generative frameworks show that ABSA subtasks can be cast as structured text generation rather than only as classifier heads [14], and text-generation formulations of aspect category sentiment analysis report particular strength in zero-shot and few-shot settings [15]. At the same time, broader text-classification evidence suggests that in-context learning remains an informative but demanding baseline: smaller fine-tuned encoders often still outperform zero-shot and few-shot prompting when labels are structured and evaluation is strict [19], and broader sentiment-analysis reviews in the LLM era recommend caution against assuming prompt-based superiority by default [30]. This literature justifies including LLM-based baselines while keeping the central contribution focused on the synthetic educational dataset and its benchmark evaluation.

The most consequential contribution is therefore the benchmark resource itself, not synthetic text generation in isolation. What makes that resource scientifically useful is the combination of three properties. First, the aspect schema is pedagogically motivated: it separates actionable dimensions of teaching and course design rather than collapsing them into overall satisfaction. Second, the generator is controlled rather than generic: target aspect sentiments and sampled nuance attributes are manipulated independently so that labels can

be realized under varied educational situations and writing styles. Third, the corpus is paired with explicit ABSA evaluation, so the study contributes not only text generation, but a reproducible experimental setting for a domain where public aspect-labeled data remain scarce.

# 3 Data Resource and Generation Protocol

The methodology has two linked components: a synthetic data generation protocol and a downstream ABSA analysis pipeline. The first produces educational reviews with controlled aspect-level sentiment labels and varied review conditions. The second evaluates whether those reviews function as useful ABSA training and evaluation data. Diversity and realism analyses are treated as quality analyses of the generator, whereas the main empirical results concern model behavior on the synthetic corpus.

## 3.1 Synthetic Data Generation Pipeline

The generation process is organized as a sequence of four decisions. First, the system samples the supervision target itself: one, two, or three aspect-sentiment pairs drawn from the 20-aspect pedagogical schema. Second, it samples a separate nuance state from the attribute families that define course context, student background, assessment conditions, writing style, and realism controls. This separation is intentional. The target labels specify what the downstream ABSA model should recover, whereas the nuance state specifies how those labels are realized in the surface review. The same aspect combination can therefore appear under different course names, student trajectories, and rhetorical styles instead of collapsing into one repeated prompt pattern.

After those two states are sampled, they are merged only at prompt construction time. A realism-constrained instruction is appended to the sampled labels and attributes to form the generation request. The generator then produces a draft review, a refinement stage removes recurrent synthetic cues while attempting to preserve the declared labels, and the exported record retains both its aspect labels and its sampled nuance attributes. Large-scale generation is therefore not driven by a single free-form prompt, but by a fixed template whose row-level variation comes from independently sampled targets and contextual controls.

Figure 1 condenses the generation protocol to its main structural idea. One input stream defines the supervision target, namely one to three aspect-sentiment labels. A second input stream defines how those labels are realized in text through sampled course context, student state, pedagogical circumstances, and style controls. These streams meet only at prompt construction time, after which generation yields a review-level benchmark record that preserves both the text and the sampled control state. The dashed feedback path is shown as an inter-cycle revision rather than a per-row operation because realism validation updates the stabilized instruction between complete prompt states.

## 3.2 ABSA Analysis Pipeline

The benchmark pipeline begins from the assembled review-level corpus and applies one shared train-validation-test split to every reported method. The train partition is used for parameter estimation, the validation partition is used for early stopping, threshold calibration, and prompt selection, and the test partition is reserved for final reporting only. This split discipline is especially important here because the paper also reports realism validation and mapped real-data transfer; those additional analyses remain outside the internal benchmark and are never merged into the synthetic split.

Within that split, the primary task formulation is two-step ABSA. A first stage predicts which aspects are present in a review, and a second stage assigns sentiment only to the aspects selected by the detector. This decomposition keeps omission errors and polarity errors analytically separate and makes the evaluation easier to interpret than a single opaque score. Reported results therefore combine multi-label detection metrics such as precision, recall, and F1 with a detected-aspect sentiment MSE that measures how well polarity is recovered once an aspect has been predicted. Prompt-based methods and mapped real-data transfer are evaluated under the same output contract, but they are reported as supporting analyses rather than as the central benchmark family.

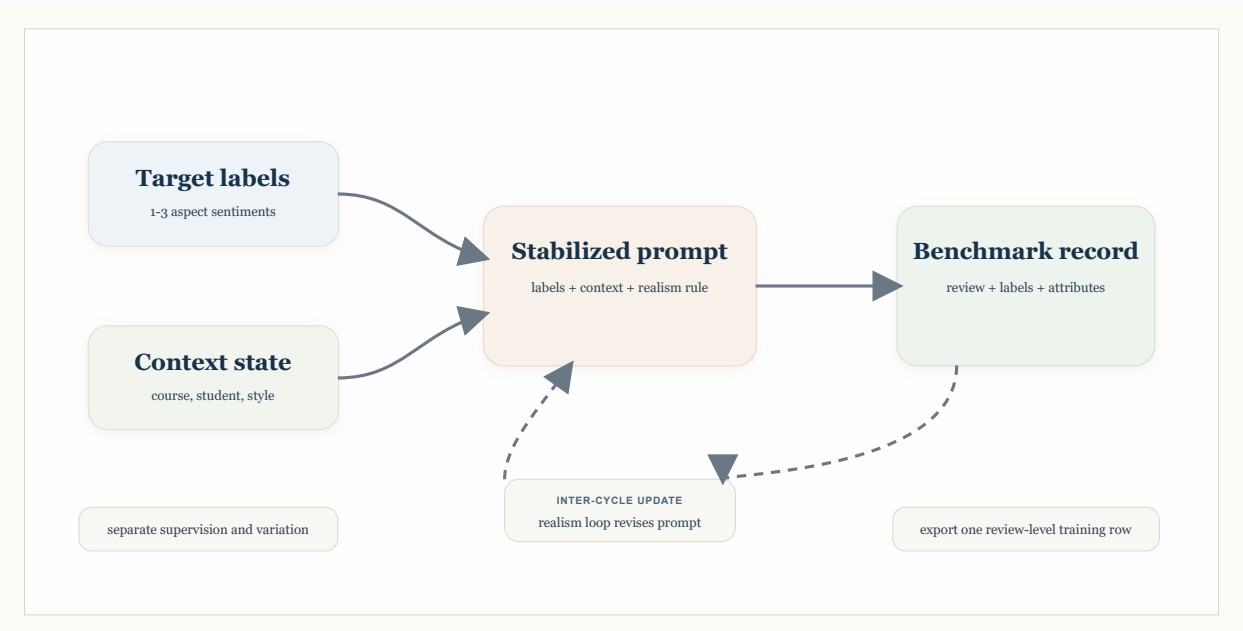

Figure 1. Supervision targets and contextual attributes are sampled separately, merged through a stabilized prompt, and exported as review-level benchmark records.

The evaluation side of the study uses a single shared contract. Every reported approach begins from the same review-level corpus and the same 8,000/1,000/1,000 split, so differences in outcome come from the modeling approach rather than from incompatible data conditions: one corpus, one split, one validation stage, and one held-out reporting stage. Validation is reserved for calibration and model choice, while mapped real-data transfer and generator-validation analyses are treated as complementary checks rather than replacements for the main synthetic benchmark.

### 3.3 Aspect Inventory and Pedagogical Scope

The benchmark uses a 20-aspect inventory so that distinct student concerns can be represented directly instead of being collapsed into broad categories such as `overall_experience` or `relevance`. For readability, the inventory is organized into five pedagogical blocks: instructional quality (`clarity`, `lecturer_quality`, `materials`, `feedback_quality`), assessment and course management (`exam_fairness`, `assessment_design`, `grading_transparency`, `organization`, `tooling_usability`), learning demand and readiness (`difficulty`, `workload`, `pacing`, `prerequisite_fit`), learning environment (`support`, `accessibility`, `peer_interaction`), and engagement and value (`relevance`, `interest`, `practical_application`, `overall_experience`). Appendix A.1 lists the full inventory with these group assignments.

These additions are motivated by pedagogy rather than by label inflation. Students frequently distinguish between a course that is difficult and one that is poorly paced, between a fair exam and a badly aligned assessment structure, and between strong content and weak tooling or feedback loops. Making those distinctions explicit is one of the main novelty claims: the goal is not only more labels, but a more educationally actionable aspect space than the small public student-feedback corpora described in prior work.

### 3.4 Prompting Protocol for Diversity

Diversity does not refer to surface paraphrasing alone. The generator varies course name, lecturer, grade, writing style, and one to three aspect labels, but that space would still be too narrow if reviews differed only superficially. The protocol therefore samples from a richer attribute schema that captures situational, pedagogical, and stylistic variation together. The schema contains four groups of controls: core context, assessment and teaching context, linguistic diversity, and realism controls.

Core-context attributes include course name, course level, semester stage, student background, motivation for taking the course, attendance pattern, study context, and grade band. Assessment-context attributes include assessment profile, instructional delivery, support-channel experience, administrative friction, feedback timing, prerequisite fit, collaboration structure, and platform or tooling experience. Linguistic-diversity attributes include writing style, emotional temperature, hedging level, specificity level, review length band, formality level, and recommendation stance. Realism-oriented attributes include review shape, contradiction pattern, time-pressure context, natural-noise style, comparison frame, and memory anchors. Each attribute is parameterized with a compact set of 4 to 6 values so the prompt remains controllable while still widening the review space. Each synthetic review samples five core-context attributes with `course_name` always required, four assessment-and-teaching attributes, three linguistic-diversity attributes, and three realism-control attributes. The result is one prompt template with a different contextual state for each review rather than a fixed bundle repeated at scale.

### 3.5 Prompt Stabilization and Data Generation Process

The generation process can be read as a separation of supervision targets from contextual realization. First, target attributes are sampled as one to three aspect-sentiment pairs; these are the labels the downstream ABSA system must recover. Second, nuance attributes are sampled independently from the four attribute groups described above so that the same target labels can appear in different pedagogical and stylistic contexts. Third, the prompt combines the sampled labels, the sampled nuance state, and the realism-constrained instruction used throughout corpus generation. The model then produces a draft review, a refinement step removes obvious synthetic cues while trying to preserve label faithfulness, and the stored item retains both its target labels and its sampled nuance attributes.

$$K \sim \text{Categorical}(0.30, 0.40, 0.30), \qquad K \in \{1, 2, 3\}$$

$$A = \{(a_i, s_i)\}_{i=1}^K, \qquad a_i \in \mathcal{A}, \quad s_i \in \{\text{negative}, \text{neutral}, \text{positive}\}$$

$$N = \bigcup_{g \in \mathcal{G}} \{(n_{g,j}, v_{g,j})\}_{j=1}^{m_g}, \qquad v_{g,j} \sim P_g(\cdot)$$

$$p_c = T(A, N, I_c), \qquad x^{(0)} \sim p_\theta(\cdot \mid p_c), \qquad x = R_\phi(x^{(0)}, A, N, I_c)$$

In this formalization, $K$ is the sampled number of aspects, $A$ is the set of target aspect-sentiment labels, $N$ is the sampled nuance state drawn from the attribute groups $\mathcal{G}$, $I_c$ is the realism-constrained instruction, $T$ is the prompt-construction function, $p_\theta$ is the base review generator, and $R_\phi$ is the refinement step that attempts to preserve labels while removing cues that repeatedly trigger synthetic detection. The notation highlights the main design choice of the paper: labels and contextual variation are sampled separately and only later combined into one review-generation prompt.

The full dataset is generated from one prompt specification rather than from mixed prompt states. That specification is rendered into OpenAI Batch requests with `gpt-5-nano` as the generator, `minimal` reasoning effort, and `low` text verbosity. Length control is enforced both textually and operationally: every prompt includes a sampled `review_length_band` attribute, the prompt injects hard length guidance for that band, and the request builder sets band-specific `max_output_tokens` budgets that further depend on whether the review has one, two, or three target aspects. The resulting corpus is therefore not produced by an unconstrained free-form prompt, but by a fixed template with per-row target labels, nuance states, and output-budget controls.

Realism is treated as more than surface messiness. The stabilized instruction explicitly discourages sentence-by-sentence checklist coverage of aspect labels, over-balanced tradeoff language, stock recommendation summaries, and generic domain-term stacking. Instead, it asks for asymmetric detail, incidental mention of some aspects, and limited but grounded specificity. The number of labeled aspects is also constrained. The synthetic corpus contains an almost uniform distribution over one-, two-, and three-aspect reviews, with counts of 2,008, 1,969, and 2,007 respectively. This empirical pattern justifies restricting generation to one to three aspects only; allowing more aspects would likely create unnaturally dense reviews and increase synthetic

detectability. The protocol therefore uses either the empirical distribution directly or a rounded practical policy of 0.30 / 0.40 / 0.30 for one, two, and three aspects respectively.

### 3.6 Synthetic Training Data and Real Validation Data

The final corpus contains 10,000 synthetic records generated from the realism-tuned prompt package. All rows contain review text and at least one valid labeled aspect, although 841 rows were marked `incomplete` by the API because they hit the output-token cap. The assembled corpus spans 8 course names, 6 observed writing-style labels, 5 grade bands, and the full 20-aspect inventory. The mean review length is 117.2 words, the median is 108 words, and each review contains an average of 2.0 labeled aspects. In practical terms, the corpus is large enough for controlled internal benchmarking, while realism validation and synthetic-to-real transfer remain separate supporting analyses.

Real data are used in two different but still narrower roles than the main synthetic benchmark. The realism-validation pool contains 32 public OMSCS reviews drawn from four course pages: `CS-6200`, `CS-6250`, `CS-6400`, and `CS-7641`. These reviews are not used for training, validation, or test evaluation in the main ABSA benchmark; they are used only as blinded reference items in the real-versus-synthetic judge loop so that prompt changes can be checked against naturally occurring educational-review language. Separately, the external transfer evaluation uses the annotated student-feedback corpus of Herath et al. [11], conservatively mapped to a 9-aspect overlap with our schema. That mapped real set is used only for out-of-domain evaluation after synthetic training and validation. This distinction is important for interpretation: synthetic data define the main ABSA benchmark, while real data are reserved for generator analysis and external evaluation.

The richness of the generator comes from the combination of target attributes and nuance attributes. Target attributes define the supervision target, while nuance attributes introduce course- and student-level variation that helps the same aspect labels appear in different contexts. In this protocol, the nuance space is grouped into four functional blocks so the prompt remains interpretable: one block anchors the course situation, one changes the pedagogical events being described, one changes linguistic realization, and one suppresses templated synthetic regularity. Table 2 lists representative variables from each block rather than exhausting the full schema, because the scientific point is the role each block plays in controlled diversity; the aspect blocks themselves are listed separately in Appendix A.1.

Table 2. Representative nuance-attribute groups used to diversify synthetic reviews while keeping the target aspect labels fixed. The table emphasizes the functional role of each block; individual reviews sample only a subset of these variables.

| Group | Representative attributes | Function in the generation protocol |
|---|---|---|
| Core context | `course_name`, `course_level`, `semester_stage`, `student_background` | Places the review in a recognizable educational situation so the same labels can appear under different student positions and course settings. |
| Assessment and teaching | `assessment_profile`, `instruction_delivery`, `support_channel_experience`, `feedback_timing` | Changes the pedagogical substance of the review by varying what the student is reacting to, not only how the review sounds. |
| Linguistic diversity | `writing_style`, `emotional_temperature`, `hedging_level`, `review_length_band` | Changes tone, compression, and rhetorical texture while preserving the declared aspect-sentiment targets. |
| Realism controls | `review_shape`, `contradiction_pattern`, `memory_anchor`, `natural_noise` | Discourages checklist-like coverage and overbalanced summaries by introducing the small irregularities common in human reviews. |

The benchmark uses the full 20-aspect inventory. The five aspect groups span instructional quality, assessment and course management, learning demand and readiness, learning environment, and engagement and value. Those dimensions are pedagogically important because they capture implementation details that often determine whether a student experience feels coherent or frustrating, whether the social learning environment is supportive, and whether course expectations match student preparation. In other words, Table 2 is not a style table; it is a control map for how pedagogical content, student position, and writing surface are disentangled during generation.

# 4 Benchmark Task and Evaluation Setup

## 4.1 Task Definition and Split Protocol

The ABSA benchmark is defined entirely on the synthetic corpus. Reviews are divided into train, validation, and test partitions using a strict three-way split. Training updates model parameters only on the training split; the validation split is reserved for early stopping, threshold calibration, and prompt-variant selection; and the test split is held out for final reporting only. This separation is central to the experimental design because the real-review pool is not part of the ABSA benchmark at all.

The experiments use the 10,000-review 20-aspect corpus with an 8,000/1,000/1,000 train-validation-test split. Metrics include per-aspect precision, recall, and F1 for aspect detection, together with sentiment mean-squared error on detected aspects. Because this is a sparse multilabel setting with many more negative than positive label decisions, the study also tracks macro balanced accuracy, macro specificity, and macro Matthews correlation as complementary diagnostics derived from the per-aspect confusion counts. Thresholds for discriminative detectors are chosen on the validation split only.

The split itself is deterministic. The benchmark harness applies a seeded permutation with seed 42, then assigns rows to train, validation, and test according to the fixed 0.80 / 0.10 / 0.10 proportions. No real-data rows are ever merged into this split. This point is worth stating explicitly because the paper contains both synthetic and real evaluations: the synthetic split is the main benchmark, the OMSCS pool is used only for realism validation, and the mapped Herath corpus is used only after training as an external evaluation set.

## 4.2 Evaluation Plan for the ABSA Pipeline

The evaluation plan compares two families of ABSA approaches under one shared task definition. The first family contains trainable encoders that learn aspect detection and per-aspect sentiment from synthetic supervision. The second contains GPT-based inference methods that recover the same outputs without task-specific fine-tuning. Both families are executed and reported in the study. The difference is not whether they are tested, but how they are deployed: the local supervised models are trained on the synthetic split, whereas the GPT-based methods perform direct batch inference on the held-out test split under the same output schema.

Formally, let $x$ denote a review, let $\mathcal{A} = \{a_1, \ldots, a_K\}$ denote the aspect inventory, let $z \in \{0, 1\}^K$ denote aspect presence indicators, and let $s \in \{-1, 0, 1\}^K$ denote aspect sentiments for the aspects that are present. The benchmark can then be viewed as learning or inferring a mapping $x \mapsto (\hat{z}, \hat{s})$, with detection quality evaluated against $z$ and sentiment quality evaluated only on detected or gold-present aspects depending on the reporting view. The main supervised decomposition treats ABSA as

$$\hat{z} = f_\theta(x), \qquad \hat{s} = g_\phi(x, \hat{z})$$

where $f_\theta$ is a multi-label detector and $g_\phi$ is a sentiment predictor applied only to the aspects selected by the first stage. This two-step formulation remains the main discriminative benchmark because it cleanly separates omission errors from polarity errors. The reported benchmark includes the TF-IDF two-step baseline together with transformer two-step models built on `distilbert-base-uncased`, `bert-base-uncased`, `albert-base-v2`, and `roberta-base`, all evaluated on the same 10K / 20-aspect train-validation-test split.

The training protocol is identical across the transformer baselines except for the encoder backbone. Reviews are truncated or padded to 192 tokens, optimization uses AdamW with learning rate $3 \times 10^{-5}$, and both the detection and sentiment stages are trained for up to three epochs with patience-based early stopping after two non-improving validation checks. The detection head is trained with sigmoid logits and a per-aspect weighted binary cross-entropy loss whose positive weights are estimated from the training split and clipped to the range $[1, 50]$ to avoid extreme imbalance. The sentiment head predicts one bounded score per aspect via a tanh output layer and is optimized with masked mean-squared error so that loss is accumulated only on gold-present aspects. This explicit protocol matters because it explains why the benchmark emphasizes reproducibility and interpretable stage-wise errors rather than aggressive architecture tuning.

Threshold calibration is also separated cleanly from test evaluation. After training the detection stage, each aspect receives its own decision threshold chosen on the validation split by a grid search from 0.05 to 0.95 in steps of 0.05, maximizing per-aspect F1 on validation only. Those fixed thresholds are then applied once on the test split. Reported micro- and macro-F1 therefore reflect calibrated binary decisions, whereas the sentiment MSE is measured only on aspects that the model actually predicts as present. The balanced-accuracy and specificity diagnostics are useful here because they remain interpretable when the label space is sparse and class prevalence differs strongly across aspects. That detected-aspect view intentionally couples polarity quality to detection quality, which makes the sentiment metric more operational but also more conservative than evaluating sentiment only on gold-present aspects.

We also retain a single-stage joint family as an explicit decomposition baseline. In that setting, one model predicts both outputs simultaneously,

$$(\hat{z}, \hat{s}) = h_\psi(x),$$

so that the benchmark can test whether jointly coupling aspect presence and polarity is competitive with the more interpretable two-step design. In the evaluation framework these joint variants are represented by compact `bert_joint` and `distilbert_joint` baselines, and they are reported as executed local comparisons alongside the stronger two-step models.

The prompt-based branch casts the same output space as structured generation rather than parameter updates. Given an instruction template $\pi$, an optional demonstration set $D$, and possibly a retrieval operator $R(\cdot)$, a prompted model produces

$$(\hat{z}, \hat{s}) = G_{\text{LLM}}(x; \pi, D), \qquad D = \varnothing \text{ for zero-shot}, \qquad D = R(x) \text{ for retrieval-based few-shot}.$$

This family includes six variants. The zero-shot model uses constrained structured decoding without demonstrations. Fixed few-shot prompting uses a static set of labeled training examples, while diverse few-shot prompting chooses examples that deliberately vary aspect count, tone, and review style. Retrieval-based few-shot prompting replaces static demonstrations with lexical nearest neighbors from the training split so that the context is review-dependent. The batch evaluation path uses a strict schema-constrained output contract, and the reported GPT results cover zero-shot, fixed few-shot, diverse few-shot, and retrieval-based few-shot inference under that shared contract.

Two additional prompt decompositions are included because they parallel common ABSA design choices. The two-pass prompted variant first predicts aspects and then conditions sentiment on those detected aspects,

$$\hat{z} = G_{\text{det}}(x; \pi_{\text{det}}, D_{\text{det}}), \qquad \hat{s} = G_{\text{sent}}(x, \hat{z}; \pi_{\text{sent}}, D_{\text{sent}}),$$

which makes its structure directly comparable to the two-step discriminative pipeline. The aspect-by-aspect prompted variant instead factors inference over the aspect inventory,

$$\hat{z}_k = G_{\text{pres}}^{(k)}(x), \qquad \hat{s}_k = G_{\text{pol}}^{(k)}(x) \text{ only if } \hat{z}_k = 1,$$

so it asks a separate binary presence question for each $a_k$ and only then queries sentiment where needed. This formulation is more expensive but useful for diagnosing whether per-aspect prompting improves recall control or sentiment calibration.

These GPT-based variants are part of the ABSA evaluation plan, not the realism protocol. The realism protocol (Appendix A.11) remains a separate judge-based validation loop used to improve the generator prompt, whereas the evaluation plan asks whether different ABSA approaches can recover the synthetic aspect labels once the dataset has been fixed. Across all planned approaches, model selection, threshold calibration, and prompt-choice decisions are confined to the validation split, and final benchmark numbers are reported only on held-out data.

Table 3 functions as a compact map of the executed experiment families. Rather than repeating implementation details, it keeps the benchmark logic visible in one place: which approaches learn from synthetic supervision, which recover labels through inference-time prompting, and which analyses are used to examine robustness and transfer.

Table 3. Benchmark matrix for the ABSA evaluation plan. The table shows the executed model families and the role each one plays in the reported evidence.

| Family | Modeling idea | Representative variants | Role in the study |
|---|---|---|---|
| Classical baseline | Two-step sparse lexical detector plus sentiment regressor | `tfidf_two_step` | Main benchmark
Low-cost reference point under the same validation and test contract. |
| Transformer encoders | Two-step supervised detection and sentiment models | `distilbert-base-uncased`, `bert-base-uncased`, `albert-base-v2`, `roberta-base` | Main benchmark
Primary discriminative comparison family on the full 10K / 20-aspect split. |
| Joint prediction | Single model predicts aspect presence and polarity together | `bert_joint`, `distilbert_joint` | Executed comparison
Used to test whether a compact joint formulation can match the stronger two-step decomposition. |
| GPT-based inference | Structured generation with demonstrations or retrieval context | `gpt-5.2` zero-shot, few-shot, few-shot-diverse, `retrieval-few-shot` | Executed comparison
Inference-only ABSA baselines under the same structured aspect-sentiment contract and the full held-out test split. |

This matrix is intentionally compact. The later results section returns to exact scores, robustness analyses, and per-aspect behavior, but Table 3 keeps the benchmark logic visible in one place: what is learned from synthetic supervision, what is inferred at test time only, and how those families complement the transfer and generator-validation analyses.

Table 4. Exact corpus-level counts and ranges for the synthetic benchmark and the two real-data pools used in realism analysis and external transfer. The table provides the precise summary values that complement the broader visual distribution shown later in Figure A7.

| Measure | Value | Interpretation |
|---|---|---|
| Clean reviews | 10,000 | Usable synthetic records in the final corpus |
| Real validation reviews | 32 | Public OMSCS reviews used only for the realism-validation loop |
| Mapped external test reviews | 2,829 | Annotated student-feedback reviews from Herath et al. used only for synthetic-to-real evaluation on a 9-aspect overlap |
| Course names | 8 | Multiple academic contexts rather than a single course |
| Observed writing styles | 6 | Style-conditioned prompting remains present in the assembled dataset |
| Aspect inventory | 20 | The benchmark runs directly on the 20-aspect pedagogical schema listed in Appendix A.1 |
| Mean / median words | 117.2 / 108 | The corpus mixes short comments with longer reflective reviews |
| Mean aspects per review | 2.0 | Many reviews are genuinely multi-aspect |
| Min-max words | 34-273 | Substantial variation in compression and detail remains visible even after length controls |

## 5 Results

The results are presented in the same order as the paper's claim structure. After mapping the reported analyses to their data and splits, we report the main local benchmark, then the full-test GPT and real-data comparisons, and only afterward return to generator-quality diagnostics, focusing on label faithfulness and a faithfulness-aware filtering recipe. This ordering is intentional: the central claim concerns the value of the synthetic corpus as an educational ABSA benchmark resource, whereas faithfulness analysis explains the strengths and limits of the generation process. The corpus profile and representative examples, corpus-scale output-control statistics, and the three-cycle realism-validation procedure are reported in full in the appendix as generator diagnostics.

### 5.1 Scope of Reported Analyses

Table 5. Evidence map for the reported analyses. Unlike Table 3, which organizes the method families, this table aligns each analysis block with its data source, split, and interpretive role in the study.

| Analysis block | Data and split | Primary outputs | Interpretive role |
|---|---|---|---|
| Synthetic benchmark | 10,000 generated reviews with an 8,000 / 1,000 / 1,000 train-validation-test split | Detection F1, sentiment MSE, runtime, per-aspect summaries, and robustness analyses | Principal evidence for learnability and model comparison on the 20-aspect benchmark. |
| GPT-based inference | The same 1,000-review held-out synthetic test split, evaluated through batch prompting | Structured sparse JSON predictions and full-test detection/sentiment metrics | Inference-time comparison under the same label contract as the learned local models. |
| External mapped evaluation | 2,829 mapped Herath student-feedback reviews over the shared 9-aspect overlap | Transfer metrics, overlap support counts, and overlap-matched synthetic-versus-real comparisons | Conservative real-data check on whether synthetic supervision connects to one annotated educational feedback space. |
| Realism study | Three 60-item cycles built from public OMSCS reviews and matched synthetic reviews | Judge accuracy, confusion, entropy, cue tags, and prompt-state transitions | Generator-side validation showing how realism cues were diagnosed and reduced across cycles. |
| Faithfulness audit | A 250-review full-corpus sample plus the 25-review pilot subset | Aspect support rates and sentiment-match rates between declared labels and textual evidence | Clarifies how the benchmark should be interpreted as a useful but imperfect supervised resource. |

Table 5 therefore complements rather than repeats Table 3. Table 3 says which method families are executed, while Table 5 says what each analysis block is meant to establish, which split it uses, and how it should be interpreted alongside the main benchmark.

### 5.2 Internal Benchmark on the 10K / 20-Aspect Corpus

The main benchmark uses the 10,000-review corpus with a strict 8,000/1,000/1,000 train-validation-test split. Seven local approaches are reported: TF-IDF, four two-step transformer encoders, and two joint encoders that predict aspect presence and polarity together. Among the untuned models, `bert-base-uncased` gives the strongest held-out detection, at macro balanced accuracy 0.623 and macro AUROC 0.681 (micro-F1 0.276, detected-aspect sentiment MSE 0.4959), followed by `distilbert-base-uncased` at balanced accuracy 0.621 (micro-F1 0.2691). The joint variants remain competitive but lower (`distilbert_joint` micro-F1 0.2524, `bert_joint` 0.2447), which supports the two-step decomposition as the clearest baseline family in this benchmark. We lead with balanced accuracy and threshold-free AUROC rather than micro-F1 because micro-F1 is a deliberately strict operating-point metric here: with one to three aspects per review among twenty mostly-absent labels, the trivial all-present baseline already scores 0.183 micro-F1, so the absolute level is compressed by the task structure and the measured label noise rather than by weak representations. Micro-F1 is retained throughout the tables as the standard comparable metric, but it is read alongside balanced accuracy and ranking quality, not alone.

Additional robustness analyses sharpen that picture. Across three seeds for TF-IDF, DistilBERT, and BERT, the highest mean detection score comes from BERT at $0.2791 \pm 0.0140$ micro-F1, while DistilBERT remains the most stable model at $0.2694 \pm 0.0005$. A modest lower-rate four-epoch BERT schedule further raises the held-out score to 0.2930 and lowers sentiment MSE to 0.4728, whereas the same change does not improve DistilBERT. Appendix A.9 reports the full seed-stability, joint-versus-two-step, and tuning tables and figures that support these observations.

Absolute scores remain moderate, which is itself informative. The benchmark covers 20 aspects across five pedagogical groups, includes subtle categories such as `feedback_quality`, `peer_interaction`, and `prerequisite_fit`, and allows one to three aspects per review across multiple writing styles and course

contexts. The ranking also has a concrete error-profile interpretation. BERT and DistilBERT improve primarily by lifting recall above 0.43 while retaining enough precision to prevent indiscriminate overprediction. ALBERT and RoBERTa, by contrast, collapse to recall 1.000 at micro-F1 0.183, and the resulting false positives inflate downstream sentiment MSE to 0.577 and 0.684 respectively. In this sense, Table 6 shows that the benchmark rewards models that recover many aspect mentions without collapsing into broad positive prediction.

Part of the moderate ceiling is a property of the labels rather than of the models. The held-out test split carries the same label faithfulness as the training corpus: under the full-corpus `gpt-4.1-mini` audit (Section 5.7), the 1,000-review held-out split has a mean per-row aspect-sentiment match rate of 0.58, statistically identical to the 0.58 of the full 10K. A held-out score is bounded by the faithfulness of the held-out labels, so a portion of the unrecovered detection F1 reflects test rows whose declared polarity is not visibly supported in the text rather than model limitation alone. This is directly measurable: re-scoring the trained `bert-base-uncased` detector on the faithful subset of the test split (the rows whose declared aspects are all supported) lifts micro-F1 from 0.277 (95% CI [0.270, 0.285]) on the full 1,000-review test set to 0.307 ([0.277, 0.337]) on the faithful subset, a paired gain of +0.030 (95% bootstrap CI [0.006, 0.054], higher in all four seeds). The model recovers labels more accurately exactly where the labels are faithful, which is the same signal the faithfulness-aware filtering in Section 5.7 exploits on the training side. The benchmark is therefore positioned as a controllable resource with an explicit, measurable label-noise level rather than as a saturated leaderboard.

Three controls establish that this signal is genuine rather than an artifact of label frequencies. First, a label-permutation control, retraining with the text-to-label correspondence destroyed, collapses detection micro-F1 to 0.182 (95% CI [0.177, 0.187]), indistinguishable from the strongest trivial baseline (predict-all-present, 0.183) and 0.094 below the real-label 0.277; the recovered F1 thus comes from text that genuinely predicts the labels. Second, detection scales monotonically with training data, from 0.183 at 250 reviews to 0.285 at 8,000, increasing in every seed with no plateau. Third, the learned model clears every trivial floor (predict-all-present 0.183, predict-by-frequency 0.000, random-at-prevalence 0.101) by at least 0.093. Consistent with the label-noise account, restricting both training and test to faithful rows raises micro-F1 to 0.319 ([0.290, 0.348], +0.043 over the full corpus) at audit row-score 1.0 and 0.340 ([0.332, 0.348], +0.064) at row-score 0.5 or above.

The additional diagnostics tell a fuller story than F1 alone. On the principal local comparison, BERT and DistilBERT are nearly tied in macro balanced accuracy at 0.6229 and 0.6207 respectively, while BERT now also holds the strongest untuned sentiment MSE and macro MCC. This is useful because the 20-aspect benchmark contains many negative label decisions per review, so balanced accuracy and specificity clarify whether a model is improving by recovering minority positives or merely by exploiting the dominant negatives. A threshold-free view confirms the same reading: on the 20-aspect held-out test set the BERT detector reaches a macro AUROC of 0.681 and a micro AUROC of 0.688, with per-aspect AUROC ranging from 0.54 to 0.83 (highest for lexically marked aspects such as `pacing` at 0.83 and `workload` at 0.81). Because AUROC is independent of the decision threshold and therefore of the noisy gold operating point, this shows the detector ranks aspect presence well above chance and that the moderate micro-F1 reflects thresholding and label noise rather than weak representations. Appendix A.10 reports these complementary diagnostics for the principal synthetic-benchmark, GPT-based, and mapped real-data comparisons.

Table 6 provides the exact values for the executed local benchmark. Figure 2 serves a different role: instead of repeating the table cell-by-cell, it visualizes the main trade-offs among detection quality, recall, runtime, and sentiment error. This separation is intentional so that the table remains the precise archival record while the figure highlights the comparative structure of the result space.

Table 6. Main held-out test results on the 10K / 20-aspect synthetic benchmark, ranked by detection micro-F1.

| Rank | Approach | Micro-F1 | Macro-F1 | Micro-recall | Sentiment MSE | Runtime (min) |
|------|----------|----------|----------|--------------|---------------|---------------|
| 1 | bert-base-uncased | 0.2760 | 0.3364 | 0.4396 | 0.4959 | 21.86 |
| 2 | distilbert-base-uncased | 0.2691 | 0.3376 | 0.4531 | 0.5044 | 15.95 |
| 3 | distilbert_joint | 0.2524 | 0.3248 | 0.4719 | 0.5428 | 6.29 |
| 4 | bert_joint | 0.2447 | 0.3208 | 0.5122 | 0.5288 | 11.58 |
| 5 | tfidf_two_step | 0.2326 | 0.2867 | 0.4595 | 0.6830 | 0.10 |
| 6 | albert-base-v2 | 0.1829 | 0.1828 | 1.0000 | 0.5773 | 22.33 |
| 7 | roberta-base | 0.1829 | 0.1828 | 1.0000 | 0.6838 | 22.19 |

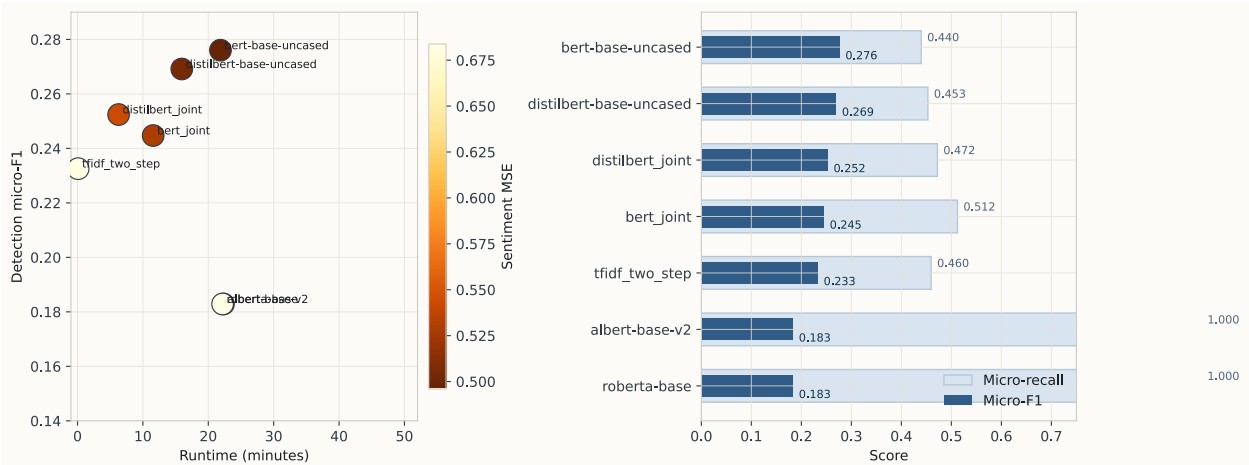

Figure 2. Local-benchmark trade-offs on the 10K / 20-aspect corpus. The left panel visualizes the cost-quality frontier using runtime, micro-F1, and sentiment MSE, while the right panel contrasts micro-F1 with micro-recall to show how some models gain coverage at the cost of more false positives.

## 5.3 GPT-Based Inference Methods

GPT-based ABSA inference is also evaluated as a full-test executed method family. Using OpenAI Batch mode with gpt-5.2, we ran zero-shot, fixed few-shot, diverse few-shot, and retrieval-based few-shot prompting over the entire 1,000-review held-out test split under the same structured output contract used throughout the benchmark. The strongest configuration is zero-shot structured prompting with a micro-F1 of 0.2519, followed closely by retrieval-based few-shot prompting at 0.2501, fixed few-shot at 0.2450, and diverse few-shot at 0.2374.

These GPT runs use the same gold label schema as the local benchmark but a different inference path. The batch job uses an exact-key sparse JSON contract so that outputs can only contain recognized aspect names and ternary sentiments, and all four variants achieved a parse success rate of 1.0 on the full test split. The zero-shot variant uses no demonstrations, the fixed few-shot variant uses three static labeled demonstrations from the synthetic training split, the diverse few-shot variant uses five demonstrations deliberately chosen to vary aspect count and tone, and retrieval-based few-shot prompting replaces static demonstrations with nearest-neighbor examples from the training split.

Table 7. Full-test GPT-based ABSA inference results under the same structured aspect-sentiment output contract used in the batch evaluation pipeline.

| Approach | Micro-F1 | Macro-F1 | Micro-recall | Sentiment MSE |
|----------|----------|----------|--------------|---------------|
| gpt-5.2 zero-shot | 0.2519 | 0.2417 | 0.3135 | 0.7179 |
| gpt-5.2 retrieval-few-shot | 0.2501 | 0.2395 | 0.3100 | 0.7244 |
| gpt-5.2 few-shot | 0.2450 | 0.2339 | 0.3045 | 0.7325 |
| gpt-5.2 few-shot-diverse | 0.2374 | 0.2261 | 0.2946 | 0.7386 |

The ranking is informative in its own right. Under this strict sparse-schema contract, zero-shot prompting and retrieval-based few-shot prompting are the strongest GPT variants, which suggests that example selection matters but that a highly constrained zero-shot formulation is already competitive. Relative to the local benchmark, the full-test GPT runs outperform TF-IDF and both joint encoders while remaining below the strongest two-step transformers. This places batch GPT inference in a meaningful middle band of the benchmark rather than at the edges of the comparison space. Because `gpt-5.2` also serves as the strict faithfulness auditor (Section 5.6) and shares a provider family with the `gpt-5-nano` generator, these are reported as same-family reference baselines rather than independent external systems. Appendix A.6 records the configuration choices that distinguish the four GPT-based inference variants.

The confusion-based diagnostics reinforce that interpretation. The strongest GPT variants reach macro balanced accuracy near 0.59 with macro specificity around 0.87, which indicates conservative but well-controlled prediction behavior: the batch prompts avoid broad overprediction, but they still miss many positive aspects that the stronger local encoders recover. In other words, GPT-based inference is competitive partly because it remains selective rather than because it dominates recall. Appendix A.10 includes these complementary values alongside the main F1-based ranking.

## 5.4 External Validation on a Mapped Real Student-Feedback Dataset

The study also includes one external validation on the annotated student-feedback dataset of Herath et al. [11]. We conservatively mapped the original annotation scheme to the subset of our pedagogical schema with defensible correspondence, yielding a 9-aspect overlap consisting of `accessibility`, `assessment_design`, `exam_fairness`, `grading_transparency`, `lecturer_quality`, `materials`, `organization`, `overall_experience`, and `workload`. This produced 2,829 mapped real reviews, which were used purely as an external evaluation set: no real reviews were used during synthetic training, validation-threshold calibration, or prompt refinement.

The transfer result is encouraging and should be read as a focused external validation. On this overlap benchmark, `bert-base-uncased` achieved the strongest detection micro-F1 at 0.4593, followed by `distilbert-base-uncased` at 0.4156 and `tfidf_two_step` at 0.3740. DistilBERT produced the lowest detected-aspect sentiment MSE at 0.3888, slightly better than BERT's 0.3990. These numbers are materially stronger than the internal 20-aspect benchmark scores, which is plausible because the external test covers fewer aspects, includes a dominant `lecturer_quality` signal, and uses a conservative overlap mapping rather than the full pedagogical schema. The result therefore supports partial synthetic-to-real transfer on one mapped educational annotation space.

To interpret that result more carefully, we also ran an overlap-matched internal comparison using the same 9 aspects on the synthetic corpus. On that overlap slice, BERT rises from 0.3869 micro-F1 internally to 0.4811 on the mapped real set, DistilBERT rises from 0.3809 to 0.4156, and TF-IDF stays roughly flat at 0.3811 versus 0.3740. This pattern indicates that the mapped real benchmark is shaped by the conservative overlap definition and by strong support for externally visible categories such as `lecturer_quality`, rather than by a uniform domain-shift penalty. Accordingly, the real-data result is best interpreted as a constrained overlap evaluation that demonstrates compatibility between the synthetic supervision scheme and one real educational annotation space. Table 9 and Figure 4 make this comparison explicit side by side, while Appendix A.4 lists the overlap counts and label balance and Appendix A.5 expands the result into a per-aspect transfer view.

The same comparison becomes more interpretable when balanced metrics are included. On the mapped real benchmark, BERT achieves the strongest macro balanced accuracy among the compared transfer models at 0.5925, slightly above DistilBERT at 0.5778 and clearly above TF-IDF at 0.5403. This supports the main transfer conclusion from another angle: the best synthetic-trained encoder is not merely matching the real overlap through dominant negative decisions, but is retaining a stronger balance between sensitivity and specificity across the nine mapped aspects. Appendix A.10 reports these values explicitly.

A real-trained reference point puts this transfer in context. Training the same `bert-base-uncased` detector on a real-Herath split and evaluating it on held-out real Herath, over the identical nine-aspect detection metric, reaches micro-F1 0.767 (95% CI [0.734, 0.800], four seeds, roughly 1,980 train / 566 test reviews). Against this same-task upper reference, the synthetic-only 0.4593, obtained with no real training data at all,

recovers about 60% of real-trained performance, which is the relevant property when no labeled institutional data are available. This is consistent with the corpus authors' own published baselines on the real data (F1 0.64 for target-oriented opinion and aspect-term extraction and 0.75 for aspect-level sentiment, using fine-tuned BERT- and RoBERTa-family encoders [11]), which use different task formulations and so serve as additional context rather than a matched comparison.

The synthetic corpus is moreover a useful initializer, not only a standalone training set. Pretraining the nine-aspect detector on the synthetic corpus and then fine-tuning it on the real-Herath train split reaches micro-F1 0.784 (95% CI [0.758, 0.809], four seeds), above the real-only baseline of 0.767 and far above the synthetic-only transfer of 0.459. Synthetic supervision therefore improves a real-data model rather than merely substituting for one, which is the property that matters for the institutional bootstrapping setting of Section 6.2: a program can pretrain on the shared synthetic corpus and fine-tune on a small adjudicated slice of its own reviews to exceed what either source supports alone.

Table 8. Exact synthetic-to-real transfer scores on the mapped Herath benchmark over the conservative nine-aspect overlap. The table gives the precise model values, while Figure 3 adds the overlap-support context needed to interpret them.

| Rank | Approach | Micro-F1 | Macro-F1 | Micro-recall | Sentiment MSE | Real reviews | Overlap aspects |
|---|---|---|---|---|---|---|---|
| 1 | bert-base-uncased | 0.4593 | 0.3059 | 0.6211 | 0.3990 | 2829 | 9 |
| 2 | distilbert-base-uncased | 0.4156 | 0.3515 | 0.6764 | 0.3888 | 2829 | 9 |
| 3 | tfidf_two_step | 0.3740 | 0.2303 | 0.4017 | 0.7019 | 2829 | 9 |

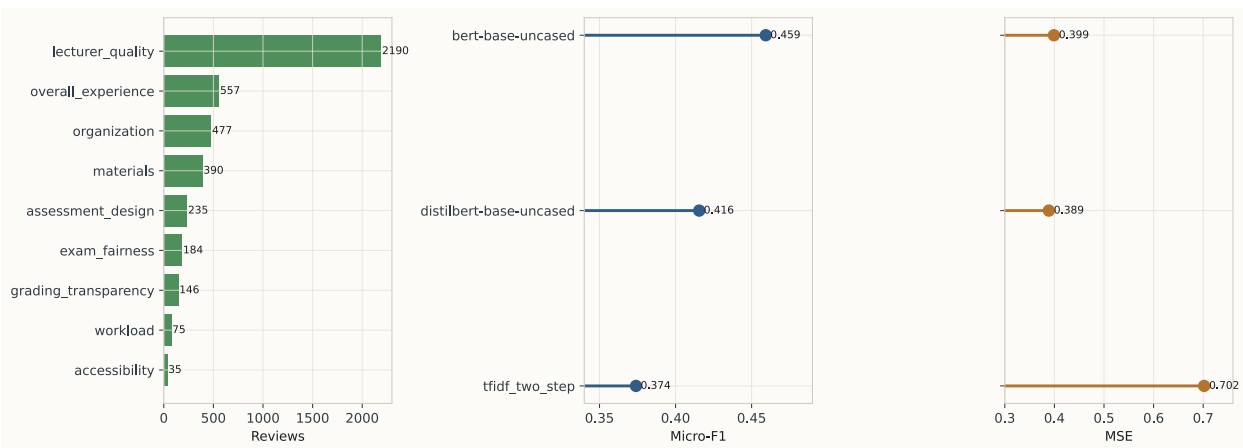

Figure 3. External validation on mapped real student feedback. The left panel provides overlap-support context from the mapped Herath corpus, while the center and right panels visualize the transfer scores reported exactly in Table 8. The figure supports a partial-transfer claim only for the conservative 9-aspect overlap.

Table 9. Side-by-side comparison on the shared nine-aspect space, contrasting the same models on the synthetic overlap test split and the mapped real-data benchmark.

| Approach | Synthetic overlap micro-F1 | Mapped real micro-F1 | Δ real minus synthetic | Synthetic overlap sentiment MSE | Mapped real sentiment MSE |
|---|---|---|---|---|---|
| bert-base-uncased | 0.3869 | 0.4811 | +0.0942 | 0.5169 | 0.3914 |
| distilbert-base-uncased | 0.3809 | 0.4156 | +0.0346 | 0.4811 | 0.3888 |
| tfidf_two_step | 0.3811 | 0.3740 | -0.0071 | 0.6519 | 0.7019 |

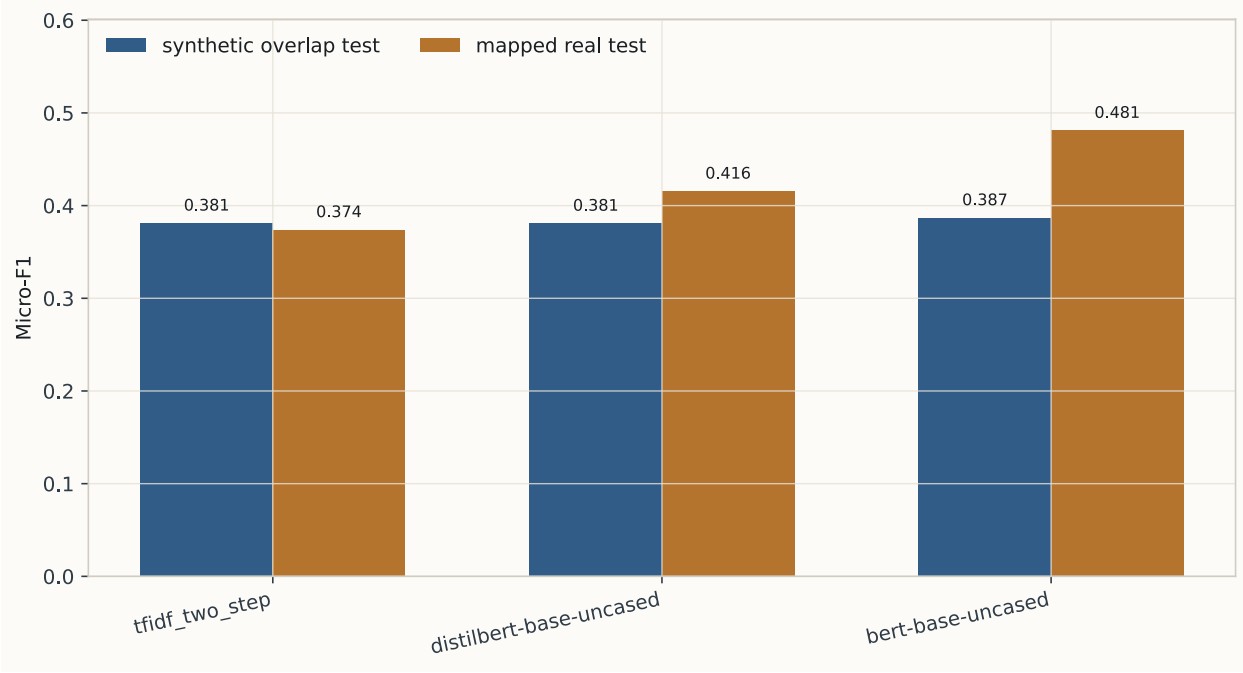

Figure 4. Overlap-matched internal versus external comparison on the same nine aspects. The absence of a uniform external drop indicates that the mapped real benchmark is partly shaped by overlap support and aspect composition rather than only by domain shift.

## 5.5 Corpus-Scale Generation Diagnostics

The next two subsections return from downstream model behavior to generator quality: first the model-assisted label-faithfulness audit, then a faithfulness-aware filtering recipe that turns that audit into a training-data selection method. Corpus-scale output-control statistics (Appendix A.14) and the three-cycle realism-validation procedure (Appendix A.13) are reported separately as generator diagnostics.

## 5.6 Model-Assisted Label-Faithfulness Audit

A complementary label-faithfulness analysis asks whether generated reviews visibly express the aspect-sentiment labels they were assigned, not only whether the text looks realistic. We therefore ran a conservative model-assisted audit with `gpt-5.2` as a strict checker over two samples: 250 reviews from the 10K corpus and all 25 reviews from the pilot sample. For each review, the auditor received the raw text and declared aspect sentiments, then judged whether each declared aspect was supported in the text and whether the polarity matched. Table 10 reports the resulting aspect-support and sentiment-match rates.

Table 10. Model-assisted label-faithfulness audit showing how often declared aspects and polarities are visibly supported in generated text.

| Split | Audit model | Reviews | Declared aspects | Aspect support rate | Aspect sentiment-match rate | Full-row support rate | Full-row sentiment-match rate |
|---|---|---|---|---|---|---|---|
| 10K corpus (250-review sample) | `gpt-5.2` | 250 | 501 | 0.7705 | 0.4232 | 0.5920 | 0.2120 |
| Pilot sample | `gpt-5.2` | 25 | 44 | 0.7727 | 0.3182 | 0.6000 | 0.2800 |

These results make the character of the corpus clearer. Many declared aspects are visibly present in the text, but polarity faithfulness is weaker, especially at full-corpus scale. At the same time, the larger 250-review audit is more favorable than the earlier smaller probe, which suggests that the benchmark preserves declared aspect presence more consistently than the harsher initial sample implied, even though polarity drift remains

substantial. The benchmark is therefore strongest as a controllable and useful educational ABSA resource whose labels remain informative but imperfect. This interpretation supports benchmarking and method comparison while also motivating future filtering or post-generation verification. Appendix A.8 retains the detailed audit table for direct comparison with later regenerated versions of the corpus.

### 5.7 Faithfulness-Aware Filtering of the Training Corpus

The 250-row audit in Section 5.6 reports a corpus-level aspect-sentiment match rate of 0.42 as a generator diagnostic. To turn that diagnostic into a method-level result, we extended the audit to the full 10,000-review corpus using a cost-matched judge (`gpt-4.1-mini`, calibrated against the `gpt-5.2` audit on the original 250 rows at per-aspect support agreement 0.874 and per-aspect sentiment-match agreement 0.790). For each row we recorded the fraction of declared aspects whose polarity is supported by the text, producing a per-row faithfulness score with a heavily quantized distribution: 22.2% of rows scored 0.00, 11.1% 0.33, 18.2% 0.50, 11.0% 0.67, and 37.5% 1.00. This per-row score averages 0.58, higher than the 0.42 aspect-sentiment-match rate reported in Section 5.6 because it is a per-row mean under the more lenient cost-matched `gpt-4.1-mini` judge rather than a per-aspect rate under the stricter `gpt-5.2` auditor; the two judges agree on 79% of per-aspect sentiment decisions.

We then partitioned the corpus into five training subsets: the highest-scoring 25% (`top25`, n = 2,500, all rows with row-score 1.00), the highest-scoring 50% (`top50`, n = 5,000), the full corpus (`full`, n = 10,000), the lowest-scoring 25% (`bot25`, n = 2,500), and a uniform-random 5,000-row sample (`random_5k`) that controls for training size. We trained BERT-base-uncased and DistilBERT-base-uncased on each bucket using the calibrated Section 5.2 recipe and evaluated each resulting model on two transfer targets: the same 9-aspect mapped Herath benchmark used in Section 5.4, and the EduRABSA 7-aspect mapping of the Hua et al. educational-review corpus (5,651 English course-and-staff reviews from RateMyProfessor, Waterloo, and Exeter). To separate effect from seed variance, every (bucket, architecture, target) configuration was retrained at eight random seeds (17, 23, 41, 42, 53, 89, 101, 137) on Modal A10G, with results aggregated across seeds.

Table 11. Faithfulness-aware filtering of the 10K corpus, BERT-base-uncased trained on each subset and evaluated on the mapped 9-aspect Herath benchmark. Mean over eight seeds with 95% bootstrap CI in brackets. Sentiment MSE is computed on aspects the detector identified as present.

| Bucket | n train | Mean row score | Sentiment MSE (detected) | Macro balanced accuracy |
|---|---|---|---|---|
| `top25` | 2,500 | 1.000 | 0.389 | 0.566 |
| | | | [0.372, 0.408] | [0.558, 0.574] |
| `top50` | 5,000 | 0.912 | **0.356** | 0.584 |
| | | | **[0.331, 0.385]** | [0.571, 0.598] |
| `full` | 10,000 | 0.577 | 0.351 | 0.580 |
| | | | [0.329, 0.376] | [0.565, 0.595] |
| `bot25` | 2,500 | 0.038 | 0.710 | 0.522 |
| | | | [0.654, 0.787] | [0.516, 0.530] |
| `random_5k` | 5,000 | 0.585 | 0.411 | 0.585 |
| | | | [0.386, 0.434] | [0.577, 0.592] |

Two findings carry the experiment, stated as paired bucket-versus-baseline differences whose 95% bootstrap CI over eight seeds excludes zero. First, at fixed training size, filtering reduces sentiment-polarity error on aspects the detector identified as present: the `top50` subset cuts sentiment MSE by 0.054 relative to the size-matched random sample (`top50 - random_5k` mean −0.054, 95% bootstrap CI [−0.090, −0.020], `top50` wins in 7 of 8 seeds). The effect replicates on DistilBERT-base-uncased at the same scale (`top50 - random_5k` mean −0.063, CI [−0.138, −0.002], 6 of 8 seeds), so the filtering signal is architecture-general rather than BERT-specific. Second, the negative-control `bot25` bucket collapses on the same metric across both architectures and both transfer targets: on BERT-Herath `bot25 - full` = +0.358 [+0.295, +0.436] (0/8 seeds), on DistilBERT-Herath +0.341 [+0.271, +0.422] (0/8), on BERT-EduRABSA +0.416 [+0.342, +0.480] (0/8), and on DistilBERT-EduRABSA +0.371 [+0.328, +0.411] (0/8). The audit score discriminates informative training rows from uninformative ones at both ends of the distribution, on both architectures, and on both transfer targets, across all 32 (bucket, architecture, target, seed) configurations of the negative

control. The size-matched filtering gain is specific to sentiment-polarity fidelity, the metric most sensitive to label faithfulness; on detection (macro balanced accuracy, Table 11) the audit score's effect is concentrated at the unfaithful end, where the `bot25` bucket alone drops to 0.522.

Because the sentiment MSE is computed only on aspects each model detected, we checked that the contrasts are not an artifact of different detection sets across buckets. They run, if anything, against the effect. At fixed training size, `top50` detects *fewer* aspects than the random sample (BERT-Herath micro-recall 0.371 vs 0.438) yet still has the lower sentiment MSE, so its win does not come from scoring an easier or larger set. The `bot25` control is the reverse: it degenerates into a high-recall, low-precision detector (micro-recall 0.793 vs the full corpus's 0.380) and nonetheless records the worst sentiment MSE. The sentiment-MSE differences therefore track label faithfulness, not how many or which aspects each model chose to score.

Table 12. Cross-architecture and cross-target replication of the Section 5.7 contrasts. Paired bucket-versus-baseline differences in sentiment MSE on the detector's predicted aspects, mean over eight seeds with 95% bootstrap CI. The `top50 - random_5k` contrast is reported on the architecture-replication row (DistilBERT-Herath) where the experiment is set up to test size-controlled filtering at the same scale as BERT-Herath; on EduRABSA the size-controlled comparison is reserved for the bot25 negative control.

| Architecture | Transfer target | top50 - random_5k | bot25 - full |
|---|---|---|---|
| BERT-base | Herath (9-aspect) | -0.054 [-0.090, -0.020] | +0.358 [+0.295, +0.436] |
| DistilBERT-base | Herath (9-aspect) | -0.063 [-0.138, -0.002] | +0.341 [+0.271, +0.422] |
| BERT-base | EduRABSA (7-aspect) | n/a | +0.416 [+0.342, +0.480] |
| DistilBERT-base | EduRABSA (7-aspect) | n/a | +0.371 [+0.328, +0.411] |

Table 12 collects these paired contrasts across both architectures and both transfer targets, and Figure 5 visualizes all conditions. Practically, the result is that the audit score in Section 5.6 is an actionable filter for downstream training. Discarding the bottom 50% of the corpus by faithfulness score matches the full-corpus sentiment-MSE level at half the training cost on Herath, and the recipe transfers to a smaller distilled architecture without loss. Discarding the bottom 25% is decisively the wrong move: that bucket is an active liability for sentiment polarity on every (architecture, target) condition we tested, with sentiment MSE 0.34 to 0.42 above the full corpus.

We additionally cross-checked that the bucketing is not specific to the chosen audit judge by re-auditing the original 250-row calibration sample with three independent cost-matched judges across two providers. Per-aspect agreement against the GPT-5.2 audit converges for all three: gpt-4.1-mini at 0.874 support / 0.790 sentiment-match, gpt-4o-mini at 0.845 / 0.715, and Claude 3.5 Haiku (via OpenRouter) at 0.817 / 0.678. The audit-derived buckets are therefore robust to the choice of cost-matched judge. Using an LLM as a label auditor is consistent with evidence that LLM annotators can rival or exceed crowd workers on annotation tasks [35]; because LLM judges also carry documented biases [36], we control for them with the cross-provider agreement here and with the human-label validation below.

Two properties make this filtering result robust to the audit being a model rather than ground truth. First, the audit score is validated behaviorally, not by judge agreement alone: the `bot25` bucket is defined purely as the rows the audit scored lowest, and training on it degrades transfer on every architecture and target above, so low-audit rows are demonstrably worse training data regardless of whether any single verdict matches a human label. Second, because the same model family both generates and audits the corpus, the score is best read as an LLM-audit signal that is useful for training-row selection rather than as a measurement of ground-truth faithfulness; the cross-provider agreement above bounds single-judge bias, and the audit additionally reproduces human faithfulness labels on annotated data, as we show next.

We close the loop by validating the audit directly against human labels. From the two human-annotated corpora (Herath and EduRABSA) we build 1,200 perturbation-controlled label variants: each sampled review carries a *faithful* variant with its human gold labels, a polarity-*flipped* variant, and an aspect-*injected* variant,

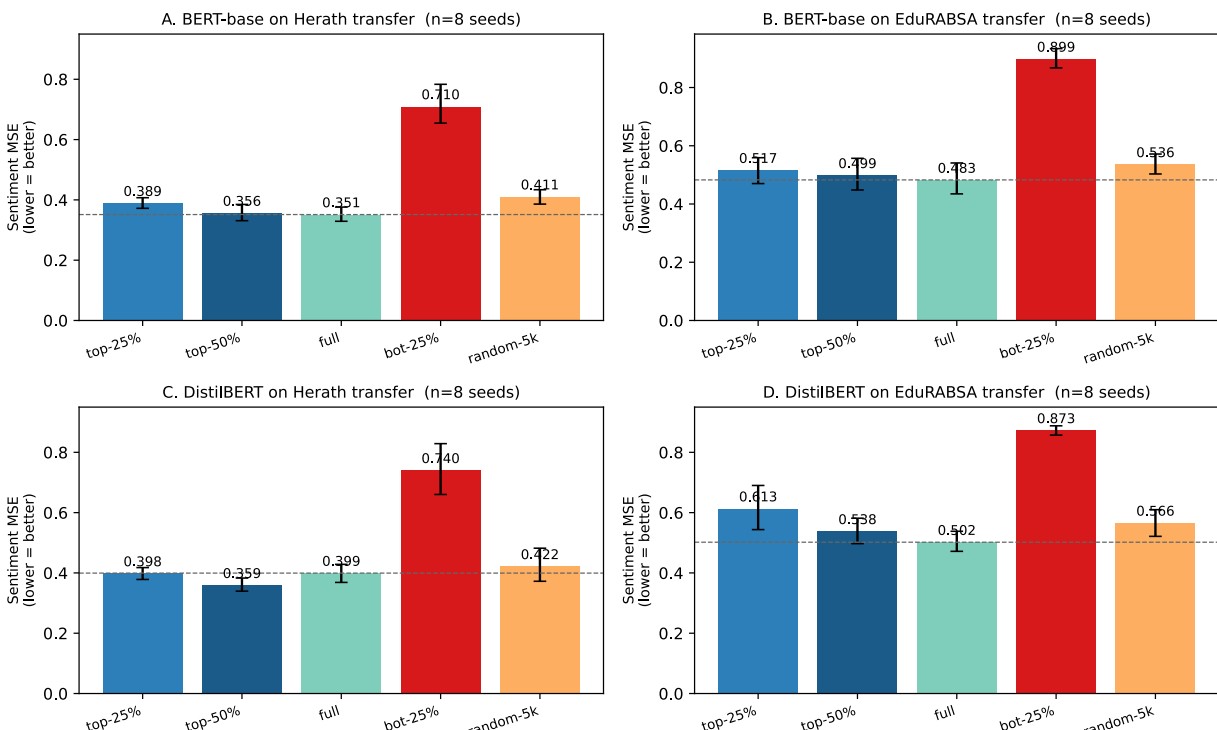

Figure 5. Faithfulness-aware filtering on the 10K synthetic corpus, evaluated across two transfer targets and two BERT-family architectures (n = 8 random seeds per condition, 95% bootstrap CI; lower is better). The `bot25` bucket fails on every condition, confirming the audit score is informative at the unfavorable end of the distribution. The `top50` bucket beats the size-matched random sample on both Herath transfer conditions, and matches the full corpus while training on half the data.

and the Section 5.6 audit scores all three. The right comparison is per aspect, because the filter audits each declared aspect independently. Across 2,482 aspect-label decisions the audit reproduces the human judgment at Cohen's kappa 0.56 (moderate agreement on the Landis-Koch scale; precision 0.88, accuracy 0.78): it retains 0.69 of genuinely faithful aspects, rejects 0.89 of polarity-flipped aspects, and rejects 0.87 of injected absent aspects, with agreement higher on EduRABSA (kappa 0.62) than on Herath (kappa 0.51). The audit that drives the filter therefore tracks human faithfulness judgments on real annotated data, not only the verdicts of other LLM judges, and its precision means a label it accepts as faithful agrees with the human annotation 88% of the time.

## 5.8 Overall Interpretation

Taken together, the study presents a coherent account of controlled synthetic supervision for educational ABSA: corpus characterization, qualitative examples, a documented ABSA modeling pipeline, calibrated 20-aspect discriminative baselines, a mapped 9-aspect synthetic-to-real evaluation on annotated student feedback, and a three-cycle realism study. The evidence supports internal learnability on synthetic reviews, informative differences among modeling choices, partial transfer to one real educational corpus, and a faithfulness-aware filtering recipe that retains the top-50% of the corpus by audit score to reduce sentiment-polarity error on transferred aspects.

# 6 Discussion

The results support interpreting the study as a synthetic educational ABSA benchmark together with a strong internal modeling analysis and one conservative external transfer check. The strongest evidence lies where those pieces connect directly: controlled generation produces aspect-labeled educational reviews, and a downstream ABSA pipeline tests whether those labels are learnable under a strict train-validation-test regime.

The study is therefore strongest as an investigation of dataset construction and internal learnability under synthetic supervision. Its value lies in offering a controllable benchmark resource for a domain where public aspect-labeled data remain scarce, together with a transparent account of how that benchmark connects to one real educational annotation space.

## 6.1 Limitations

Five limitations remain central. First, external validation uses two real educational corpora (Herath 9-aspect, EduRABSA 7-aspect) and conservative aspect overlaps, so broader claims about the entire 20-aspect schema still require additional real-data tests. Second, the realism-validation analysis is conducted with LLM judges and improves prompt quality without establishing statistical indistinguishability from real reviews; the label-faithfulness audit is validated three ways (a behavioral negative control, cross-provider judge agreement, and direct agreement with human labels at Cohen's kappa 0.56 on two annotated corpora, Section 5.7), and human re-annotation across the full 20-aspect schema remains the natural extension. Third, the 10K dataset is usable but not perfectly controlled, because a nontrivial share of rows reached the output-token cap and weakened length-band adherence at scale. Fourth, downstream training on the 10K corpus benefits from the Section 5.7 top-50% faithfulness filter (validated across two BERT-family architectures and both transfer targets); the audit score is the validated way to select training rows for sentiment-polarity fidelity on transferred aspects. Fifth, the GPT-based comparison now covers a full-test `gpt-5.2` benchmark, but it still represents one provider model and one structured prompting family rather than the full space of proprietary LLM baselines.

These limitations help define the next stage of work rather than undercut the contribution of the study. The reported evidence supports a resource-and-benchmark study with meaningful generator validation, informative internal comparisons, and a first mapped real-data check.

The most important extensions are broader synthetic-to-real tests across more real educational datasets and more of the 20-aspect schema, improved generation or filtering to raise label faithfulness and sentiment correctness on generated reviews, and broader prompt-based comparisons that add decomposition variants and cross-provider LLM baselines on the same fixed benchmark split.

## 6.2 Educational Implications

Although this study is framed as a benchmark-and-corpus contribution, the underlying motivation is practical: institutions accumulate large volumes of course feedback, and aspect-level summaries are pedagogically more useful than overall sentiment. A controllable synthetic resource has four complementary roles in this setting, each tied to a concrete institutional workflow and to the constraints that the label-faithfulness audit makes visible.

First, it lowers the cost of *model bootstrapping.* Producing the per-comment annotations needed for an institution-specific ABSA model is expensive: even at modest academic-crowdsourcing rates, a multi-thousand-comment private corpus typically costs several thousand dollars in annotation plus reviewer adjudication, often beyond the budget of a single department. A 10,000-review synthetic corpus generated through an LLM API is a one-time generation cost far below that annotation budget and is reusable across institutions. Programs that cannot fund private annotation can pretrain on the synthetic corpus and fine-tune on a small adjudicated slice of their own reviews, capturing the bulk of supervision at a fraction of the cost. The shared schema also avoids the situation in which each institution invents a new, incompatible aspect inventory, which has historically made cross-institution comparison impossible.

Second, it supports *longitudinal monitoring.* Course-improvement cycles depend on comparable aspect distributions across terms, instructors, and cohorts. A department running the same course over eight semesters cannot reliably interpret year-over-year shifts in raw sentiment without a stable instrument: shifts may reflect rater drift, syllabus changes, or external events, and the instrument itself may have drifted between iterations. A documented synthetic benchmark gives the modelling layer a stable yardstick against which a department-specific model can be tracked; the benchmark fixes the schema, the splits, and the evaluation, so when a per-aspect F1 changes on the department's own data, the change is attributable to the data rather than the instrument.

Third, it allows *controlled stress-testing.* The synthetic corpus exposes minority aspects (`accessibility`, `grading_transparency`, `peer_interaction`) that are rarely well represented in any single institution's archive. A program with two semesters of real comments has effectively no signal on `accessibility`; the synthetic corpus has hundreds of items per minority aspect, so it can act as a probing set for known coverage gaps. Institutions can thereby diagnose whether their model handles low-frequency but high-stakes aspects before deploying it on a real cohort, which is particularly relevant for accessibility and grading-transparency aspects whose mishandling has direct equity implications.

Fourth, it supports *instructor-level reflective practice.* Aspect-tagged feedback enables individualised end-of-term debriefs (a mathematics instructor receives a summary highlighting `clarity` and `pacing` trends across their own sections rather than program-level averages) and feeds faculty-development workshops with concrete patterns rather than anecdotes. Used this way, the model becomes part of formative review rather than summative evaluation, and noise on individual labels matters less than the aggregate pattern across many comments.

None of these uses presume that synthetic supervision replaces human annotation. The label-faithfulness audit constrains them directly: with overall aspect-sentiment match at 0.42 on a 250-review sample, the corpus should be treated as a noisy training resource rather than as a calibrated ground truth. Practical guidance follows. For the model-bootstrapping and stress-testing workflows, the noise is tolerable because outputs are aggregated and the resource serves model development rather than the final classifier; the institution always fine-tunes on local data before deployment. For the longitudinal-monitoring and reflective-practice workflows, deployments built on top of the corpus should be checked against locally adjudicated examples, and the faithfulness-aware filtering result reported in Section 5.7 should be applied: retaining the top 50% of the corpus by audit score reduces sentiment-polarity error on transferred aspects at half the training cost. In governance terms, the resource is appropriate for low-stakes, aggregate, formative-feedback workflows and inappropriate for high-stakes personnel decisions about instructors without human-in-the-loop review, a documented appeals process, and local institutional approval.

## 6.3 Ethics Statement

The synthetic corpus released with this study contains no identifiable student data: instructor names, course codes, and personas are sampled or invented as part of the controlled-generation protocol, and the corpus is generated rather than scraped. The mapped external evaluation uses the publicly released Herath et al. 2022 student-feedback corpus under its MIT license; that release was prepared by the authors of the original study with their institutional approvals, and our use is limited to evaluation, citation, and a conservative schema mapping documented at `paper/real_transfer/herath_mapping.json`. Preliminary realism-validation work also drew on the public OMSCS course-review pages; only public text was used and no individual reviewer is identified in any downstream artifact. Any institutional deployment of the resulting models should re-consent or re-anonymize student-authored text per local research-ethics policy and should make clear to students and instructors that an automated aspect summary is produced from feedback they submit. We do not recommend using model outputs as inputs to high-stakes personnel decisions about instructors without human review.

# 7 Conclusion

This study contributes a 10K synthetic educational ABSA corpus over a 20-aspect pedagogical schema, a documented generation protocol, and a benchmark setting that makes internal train-validation-test evaluation possible in a domain where public labeled data remain difficult to obtain. It also includes a conservative synthetic-to-real evaluation on mapped annotated student feedback, which supports partial transfer on a 9-aspect overlap, and generator validations that clarify how realism control and label faithfulness behave at corpus scale. Taken together, the evidence supports controlled synthetic supervision as a productive way to study educational ABSA, compare model families, and establish an openly reusable benchmark in a data-scarce domain.

Reproducibility: the generation artifacts, local benchmark outputs, and realism-study files are kept distinct throughout, so each reported number traces to its own artifact.

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

## A   Appendix

### A.1   Twenty-Aspect Inventory

Table A1 lists the 20-aspect inventory used for generation, realism validation, and benchmarking, grouped into the five pedagogical blocks discussed in Sections 3 and 6.

### A.2   Side-by-Side Real and Synthetic Examples

Table A2 shows representative real and synthetic review pairs from the realism study. For the synthetic reviews, the table includes both the target aspect labels and the sampled nuance attributes used to construct the prompt.

Table A1. Twenty-aspect pedagogical inventory used throughout generation, validation, and benchmarking, grouped into five pedagogical blocks.

| Group | Aspect | Description |
|---|---|---|
| Instructional quality | clarity | How understandable the teaching and explanations feel. |
| Instructional quality | lecturer_quality | Perceived quality of the lecturer or lead instructor. |
| Instructional quality | materials | Usefulness of slides, notes, readings, and resources. |
| Instructional quality | feedback_quality | Usefulness and timeliness of feedback on student work. |
| Assessment and course management | exam_fairness | Whether exams feel aligned and fair. |
| Assessment and course management | assessment_design | Alignment and structure of assignments, projects, and exams. |
| Assessment and course management | grading_transparency | How clearly grading criteria, rubrics, and score interpretation are communicated. |
| Assessment and course management | organization | Administrative clarity, course structure, and coordination. |
| Assessment and course management | tooling_usability | Friction or support created by LMS, submission systems, and required software. |
| Learning demand and readiness | difficulty | Conceptual or technical challenge of the course. |
| Learning demand and readiness | workload | Amount of sustained effort required across the term. |
| Learning demand and readiness | pacing | Whether the course tempo and weekly rhythm are manageable. |
| Learning demand and readiness | prerequisite_fit | How well the course matches the advertised prerequisite level and student preparation. |
| Learning environment | support | Quality of help from instructor, TAs, or forums. |
| Learning environment | accessibility | Perceived accessibility and inclusiveness of materials, pace, and course participation. |
| Learning environment | peer_interaction | Whether peer discussion, teamwork, and class community help or hinder learning. |
| Engagement and value | relevance | Perceived usefulness to the program or future goals. |
| Engagement and value | interest | Level of engagement or curiosity the course creates. |
| Engagement and value | practical_application | Connection to real-world practice or authentic tasks. |
| Engagement and value | overall_experience | Global student impression after tradeoffs. |

Table A2. Representative real and synthetic review pairs from the realism study. Each pair is grouped into a short real row and a matched synthetic row so the review text remains readable while the synthetic example still exposes its label and prompt context.

| Source | Review text | Annotation context |
|---|---|---|
| Pair 0 | | |

Table A2. Representative real and synthetic review pairs from the realism study. Each pair is grouped into a short real row and a matched synthetic row so the review text remains readable while the synthetic example still exposes its label and prompt context.

| Source | Review text | Annotation context |
|---|---|---|
| Real | | Public real review used only as a blinded realism-validation item. |
| | `I took this class my first semester`
`↪  in the program, alongside`
`↪  networks (6250). I almost feel`
`↪  like I shouldn't review this -- I`
`↪  didn't take advantage of the`
`↪  resources it offered. The`
`↪  instructor and TAs all seemed`
`↪  great and response time to`
`↪  questions and concerns was near`
`↪  immediate. But I acted like they`
`↪  didn't exist except the few times`
`↪  I had a specific question, and I`
`↪  suspect my learning suffered for`
`↪  it. The lectures comprise the`
`↪  bulk of the learning material,`
`↪  along with journal articles` | |
| Synthetic | `↪  assigned for most lectures.` | Target aspects |
| | `CS-6200 / Graduate Introduction to`
`↪  Operating Systems was useful for`
`↪  me, but honestly not a fun class.`
`↪  I took it my last semester before`
`↪  graduation and it definitely felt`
`↪  like one of those mid-program`
`↪  core courses that everyone should`
`↪  probably do anyway. What I`
`↪  remember most is one systems`
`↪  assignment week where I was up`
`↪  way too late debugging something`
`↪  dumb, and the peer chat helped`
`↪  more than the actual course`
`↪  materials.` | `{"lecturer_quality": "negative", "relevance":`
`↪  "positive"}`

Sampled nuance attributes

`{"course_name": "Graduate Introduction to`
`↪  Operating Systems", "lecturer_name":`
`↪  "Prof. Alvarez", "course_level":`
`↪  "mid-program core course",`
`↪  "semester_stage": "last semester before`
`↪  graduation", "attendance_pattern": "used`
`↪  the materials selectively when needed",`
`↪  "collaboration_structure": "peer`
`↪  discussion helps more than the official`
`↪  materials", "platform_and_tooling": "the`
`↪  LMS and submission flow are smooth",`
`↪  "prerequisite_fit": "starts accessible`
`↪  but ramps up sharply",`
`↪  "support_channel_experience": "email`
`↪  support is slow when deadlines hit"}` |
| Pair 1 | | |
| Real | | Public real review used only as a blinded realism-validation item. |
| | `TLDR: This was a great course, got a`
`↪  B overall but thoroughly enjoyed`
`↪  it. This course requires a lot of`
`↪  time and prerequisite knowledge`
`↪  of C and C++. Exams are`
`↪  disconnected from the project`
`↪  work. 25+ hours/week. This was my`
`↪  first class in OMSCS. I wanted a`
`↪  challenge and I definitely got`
`↪  it.` | |

Table A2. Representative real and synthetic review pairs from the realism study. Each pair is grouped into a short real row and a matched synthetic row so the review text remains readable while the synthetic example still exposes its label and prompt context.

| Source | Review text | Annotation context |
|---|---|---|
| Synthetic | | Target aspects |
| | `Honestly the platform stuff in`
`↪  CS-6200 was a pain and caused way`
`↪  more stress than it needed to. I`
`↪  took Graduate Introduction to`
`↪  Operating Systems mostly because`
`↪  it fit my schedule, not because`
`↪  OS was my thing, and that`
`↪  probably showed. I have a pretty`
`↪  strong coding background, but the`
`↪  theory side was rough for me, so`
`↪  I kept falling behind and then`
`↪  basically crammed around project`
`↪  deadlines.` | `{"interest": "positive", "accessibility":`
`↪  "negative"}`

Sampled nuance attributes

`{"course_name": "Graduate Introduction to`
`↪  Operating Systems", "student_background":`
`↪  "strong coding background but weak`
`↪  theory", "attendance_pattern": "fell`
`↪  behind and caught up around deadlines",`
`↪  "course_level": "advanced elective",`
`↪  "motivation_for_taking_course": "fit the`
`↪  schedule more than the interests",`
`↪  "platform_and_tooling": "platform quirks`
`↪  create avoidable stress",`
`↪  "feedback_timing": "scores arrive fast`
`↪  but explanations are thin",`
`↪  "collaboration_structure": "collaboration`
`↪  expectations feel unclear"}` |

## A.3 Prompt Instructions Across Realism-Validation Cycles

The realism-validation procedure rewrites the prompt between complete 60-question cycles. The prompt used in the next cycle is the editor-refined output from the previous cycle when correctly detected synthetic reviews provide sufficient cue evidence. Table A3 shows the generation instruction at each cycle.

Table A3. Prompt instructions across realism-study cycles, showing how the generation instruction changed as synthetic cues were identified and reduced.

| Stage | Prompt instruction |
|---|---|
| **Baseline cycle-0 instruction** | `Write a realistic first-person course review with uneven detail. Avoid`
`↪  textbook sentiment wording, avoid obvious label leakage, and keep the tone`
`↪  consistent with the sampled student persona.` |
| **Cycle-1 stable instruction** | `Write a realistic first-person course review with uneven detail and a mildly`
`↪  informal voice. Include 1-2 concrete, course-plausible specifics (for`
`↪  example a project, tool, rubric quirk, deadline pattern, exam format, or`
`↪  memorable incident), but do not force every aspect to appear. Avoid`
`↪  textbook sentiment wording, explicit label leakage, neat pros/cons`
`↪  symmetry, and generic praise/complaint lists. Let the review sound a`
`↪  little partial or messy, like recalled experience rather than a polished`
`↪  summary, and end naturally with a complete final thought. Keep all details`
`↪  consistent with the sampled student persona and the actual`
`↪  course/instructor context.` |

Table A3. Prompt instructions across realism-study cycles, showing how the generation instruction changed as synthetic cues were identified and reduced.

| Stage | Prompt instruction |
|---|---|
| **Cycle-2 stable instruction** | |
| | ```
Write a realistic first-person course review in a mildly informal voice. Make
↪  it feel like recalled experience, not a balanced evaluation: focus on 1-2
↪  things the student would actually remember, with uneven detail and some
↪  partialness. Include at most 1-2 concrete, course-plausible specifics
↪  (such as a project, tool, grading quirk, deadline pattern, exam format, or
↪  small incident), and make at least one of them individualized rather than
↪  just subject jargon. Do not try to cover every aspect or balance praise
↪  and criticism; avoid tidy contrast patterns, stacked common review motifs,
↪  generic domain-term lists, and polished summary phrasing. Keep the
↪  sentiment and details consistent with the sampled student persona and the
↪  real course/instructor context, and end with a natural complete final
↪  sentence.
``` |
| **Final generation instruction** | ```
You are writing one realistic student course review for research validation.
The review must feel like a naturally written student comment rather than a
↪  labeled synthetic sample.
Target aspect sentiments:
{aspect_block}
Target attributes:
{attribute_block}
Requirements:
- Keep the review first-person and specific.
- Do not mention aspect labels or sentiment labels explicitly.
- Do not force a tidy conclusion.
- Do not cover every aspect with the same level of detail.
- Let at least one point feel incidental rather than checklist-driven.
- Preserve mixed feelings when the attributes imply them.
- Additional stable realism instruction: Write a realistic first-person course
↪  review in a mildly informal voice. Make it feel like recalled experience,
↪  not a balanced evaluation: focus on 1-2 things the student would actually
↪  remember, with uneven detail and some partialness. Include at most 1-2
↪  concrete, course-plausible specifics (such as a project, tool, grading
↪  quirk, deadline pattern, exam format, or small incident), and make at
↪  least one of them individualized rather than just subject jargon. Do not
↪  try to cover every aspect or balance praise and criticism; avoid tidy
↪  contrast patterns, stacked common review motifs, generic domain-term
↪  lists, and polished summary phrasing. Keep the sentiment and details
↪  consistent with the sampled student persona and the real course/instructor
↪  context, and end with a natural complete final sentence.
Return only the review text.
``` |

## A.4 Mapped Real-Data Overlap and Label Balance

Table A4 documents the conservative overlap used for external validation against the Herath student-feedback corpus. The mapped real benchmark is intentionally narrower than the full 20-aspect schema, because only defensible correspondences were retained.

Table A4. Exact support and polarity balance for the nine overlap aspects retained in the mapped Herath benchmark. This appendix table preserves the full counts behind the lighter overlap context summarized in Figure 3.

| Aspect | Reviews | Positive | Neutral | Negative |
|---|---|---|---|---|
| lecturer_quality | 2190 | 1402 | 398 | 390 |
| overall_experience | 557 | 395 | 61 | 101 |
| organization | 477 | 266 | 155 | 56 |
| materials | 390 | 137 | 194 | 59 |
| assessment_design | 235 | 83 | 100 | 52 |
| exam_fairness | 184 | 29 | 97 | 58 |
| grading_transparency | 146 | 49 | 66 | 31 |
| workload | 75 | 12 | 21 | 42 |
| accessibility | 35 | 11 | 12 | 12 |

## A.5  Per-Aspect Synthetic-to-Real Diagnostics

Figure A1 expands the external-validation result into an aspect-by-model matrix. It is included in the appendix because it is diagnostically valuable, but too detailed for the main body.

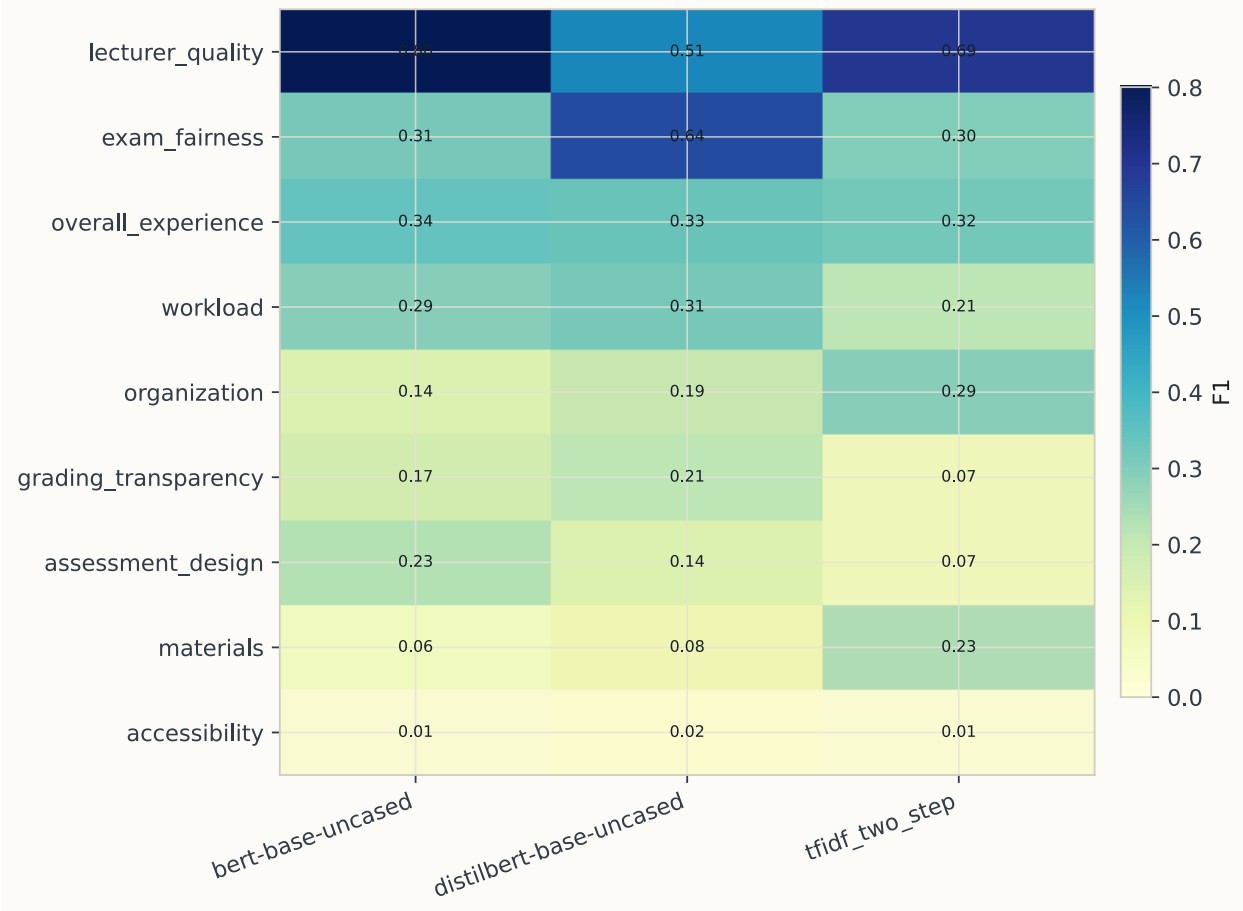

Figure A1. Per-aspect detection F1 for each synthetic-trained model on the mapped real benchmark. The heatmap shows that cross-dataset transfer is highly uneven: `lecturer_quality` and `exam_fairness` transfer more cleanly than sparse or weakly aligned categories such as `accessibility`.

### A.6  GPT-Based Inference Configuration Details

Table A5 complements Table 7 by documenting the configuration differences behind the four GPT-based inference variants rather than repeating the same held-out scores. All four methods used the same exact-key sparse JSON contract and achieved parse success of 1.00 on the full 1,000-review test split.

Table A5. Configuration details for the GPT-based inference methods reported in Table 7.

| Approach | Demonstration policy | Context selection | Test reviews | Parse success | Intended comparison |
|---|---|---|---|---|---|
| `gpt-5.2` `zero-shot` | No demonstrations | Schema instructions only | 1000 | 1.00 | Measures the strongest constrained inference baseline without example conditioning. |
| `gpt-5.2` `retrieval-` `few-shot` | Nearest-neighbor labeled demonstrations | Examples retrieved from the synthetic training split per test review | 1000 | 1.00 | Tests whether instance-level example selection improves over fixed prompting. |
| `gpt-5.2` `few-shot` | Three static labeled demonstrations | Fixed examples drawn from the synthetic training split | 1000 | 1.00 | Reference few-shot condition with stable prompt context across the full test set. |
| `gpt-5.2` `few-shot-` `diverse` | Five static demonstrations with varied tone and aspect count | Curated fixed examples from the synthetic training split | 1000 | 1.00 | Tests whether broader example diversity changes the precision-recall trade-off. |

### A.7  Pilot Subset Procedure-Validation Details

The pilot subset is included to document the small-scale validation used before full-scale generation, even though it is far too small for substantive model comparison. Table A6 records the acceptance criteria and Figure A2 shows the sampled aspect-count mix used in that validation subset.

Table A6. Pilot-subset validation criteria used before full-scale generation.

| Criterion | Value | Interpretation |
|---|---|---|
| Pilot subset | 25 reviews | Small-scale generation used for end-to-end validation of the data contract |
| Split sizes | 21 / 2 / 2 | Train / validation / test |
| Completed rate | 1.00 | All pilot responses completed |
| Text success rate | 1.00 | All rows returned parsable review text |
| Duplicate rate | 0.00 | No duplicate reviews detected |
| Length-band match rate | 0.80 | Length control cleared the predeclared pilot threshold |
| Mean review length | 125.6 words | Observed average output length under the validated generation settings |

### A.8  Label-Faithfulness Audit Details

Table A7 records the strict model-assisted label-faithfulness audit used to check whether declared aspect sentiments are visibly supported in the generated text. The main body reports the headline implication; this appendix keeps the exact audited rates visible for future comparison after regeneration or filtering.

Table A7. Detailed label-faithfulness audit rates for the full-corpus sample and the pilot sample.

| Split | Audit model | Reviews | Declared aspects | Aspect support rate | Aspect sentiment-match rate | Full-row support rate | Full-row sentiment-match rate |
|---|---|---|---|---|---|---|---|
| 10K corpus (250-review sample) | `gpt-5.2` | 250 | 501 | 0.7705 | 0.4232 | 0.5920 | 0.2120 |
| Pilot sample | `gpt-5.2` | 25 | 44 | 0.7727 | 0.3182 | 0.6000 | 0.2800 |

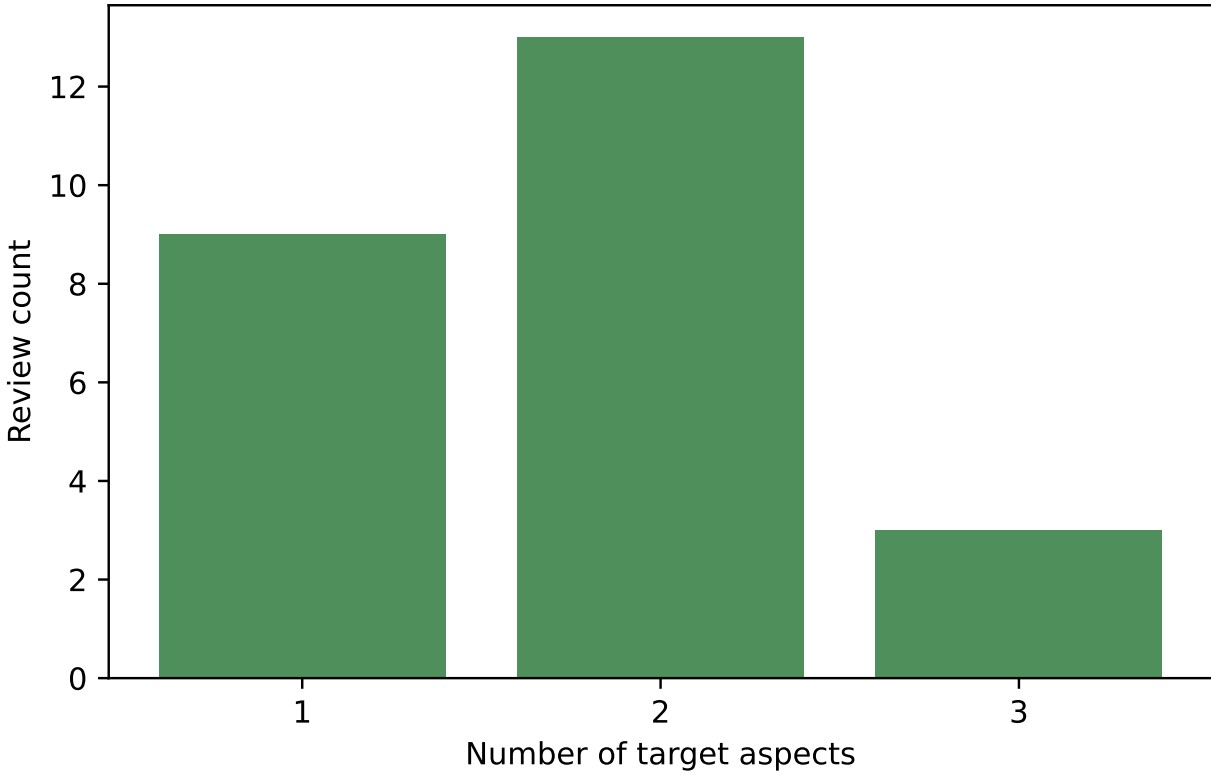

Figure A2. Distribution of sampled target-aspect counts in the pilot validation subset. This appendix figure is retained because it documents the pilot mixture without duplicating the acceptance criteria already summarized in Table A6.

### A.9 Additional Local-Benchmark Robustness Analyses

Tables A8-A10 and Figures A3-A4 collect the robustness analyses cited in Section 5.2. They show how the two-step and joint formulations compare directly, how stable the strongest local models are across three seeds, and how a modest training-budget change affects BERT and DistilBERT.

Table A8. Joint-versus-two-step comparison on the held-out synthetic benchmark.

| Family | Approach | Micro-F1 | Macro-F1 | Micro-recall | Sentiment MSE | Runtime (min) |
|---|---|---|---|---|---|---|
| Two-step | `bert-base-uncased` | 0.2760 | 0.3364 | 0.4396 | 0.4959 | 21.86 |
| Two-step | `distilbert-base-uncased` | 0.2691 | 0.3376 | 0.4531 | 0.5044 | 15.95 |
| Joint | `distilbert_joint` | 0.2524 | 0.3248 | 0.4719 | 0.5428 | 6.29 |
| Joint | `bert_joint` | 0.2447 | 0.3208 | 0.5122 | 0.5288 | 11.58 |

Table A9. Three-seed stability summary for the strongest local benchmark models.

| Approach | Seeds | Micro-F1 mean | Micro-F1 std | Sentiment MSE mean | Sentiment MSE std |
|---|---|---|---|---|---|
| `bert-base-uncased` | 3 | 0.2791 | 0.0140 | 0.5004 | 0.0296 |
| `distilbert-base-uncased` | 3 | 0.2694 | 0.0005 | 0.5097 | 0.0188 |
| `tfidf_two_step` | 3 | 0.2353 | 0.0058 | 0.6667 | 0.0158 |

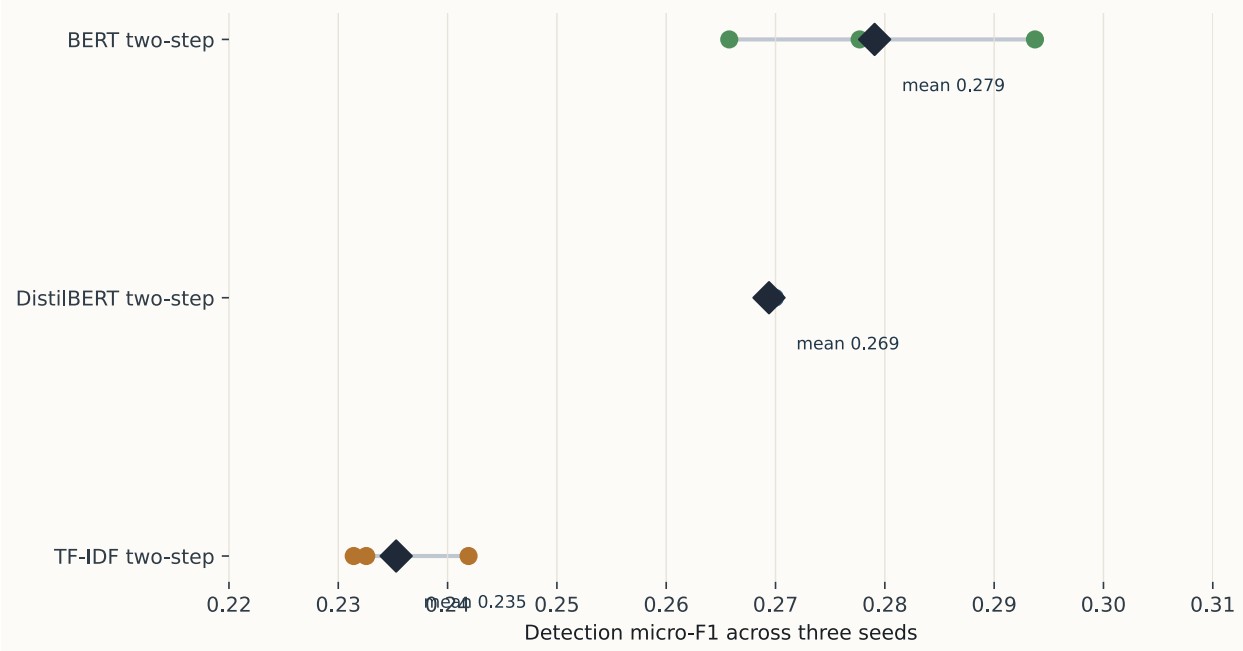

Figure A3. Micro-F1 across three random seeds for TF-IDF, DistilBERT, and BERT. The figure highlights the strong stability of DistilBERT and the larger upside variance of BERT.

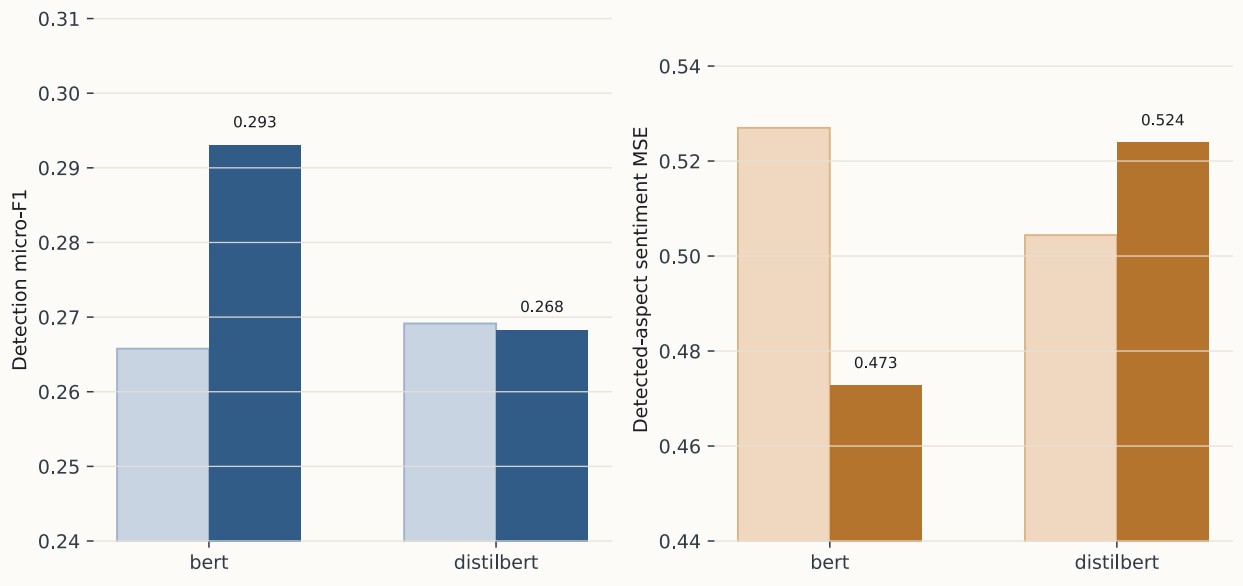

Figure A4. Effect of a modest training-budget change on the two strongest transformer baselines. BERT benefits noticeably from the longer lower-rate schedule, while DistilBERT remains essentially flat.

Table A10. Targeted training-budget comparison for the two strongest transformer baselines.

| Approach | Baseline micro-F1 | Tuned micro-F1 | Δ micro-F1 | Baseline sentiment MSE | Tuned sentiment MSE | Δ sentiment MSE |
|---|---|---|---|---|---|---|
| `bert-base-uncased` | 0.2760 | 0.2930 | +0.0170 | 0.4959 | 0.4728 | -0.0231 |
| `distilbert-base-uncased` | 0.2691 | 0.2683 | -0.0009 | 0.5044 | 0.5239 | +0.0195 |

## A.10 Additional Detection Diagnostics

Table A11 complements the headline F1 metrics with three confusion-based summaries that are less sensitive to label sparsity: macro balanced accuracy, macro specificity, and macro Matthews correlation. These values are especially useful in the present 20-aspect multilabel setting because they show whether a model is improving through better minority-positive recovery, through conservative rejection of negatives, or through a more balanced combination of both.

Table A11. Additional confusion-based detection diagnostics for the principal synthetic-benchmark, GPT-based, and mapped real-data comparisons.

| Family | Approach | Micro-F1 | Macro-F1 | Macro balanced accuracy | Macro specificity | Macro MCC | Sentiment MSE |
|---|---|---|---|---|---|---|---|
| Local synthetic benchmark | `bert-base-uncased` | 0.2760 | 0.3364 | 0.6229 | 0.8050 | 0.2766 | 0.4959 |
| Local synthetic benchmark | `distilbert-base-uncased` | 0.2691 | 0.3376 | 0.6207 | 0.7863 | 0.2713 | 0.5044 |
| Local synthetic benchmark | `tfidf_two_step` | 0.2326 | 0.2867 | 0.5955 | 0.7225 | 0.1920 | 0.6830 |
| GPT batch inference | `gpt-5.2` zero-shot | 0.2519 | 0.2417 | 0.5899 | 0.8686 | 0.1799 | 0.7179 |
| GPT batch inference | `gpt-5.2` retrieval-few-shot | 0.2501 | 0.2395 | 0.5883 | 0.8693 | 0.1823 | 0.7244 |
| GPT batch inference | `gpt-5.2` few-shot | 0.2450 | 0.2339 | 0.5848 | 0.8679 | 0.1798 | 0.7325 |
| GPT batch inference | `gpt-5.2` few-shot-diverse | 0.2374 | 0.2261 | 0.5800 | 0.8673 | 0.1653 | 0.7386 |
| Mapped real-data transfer | `bert-base-uncased` | 0.4593 | 0.3059 | 0.5925 | 0.8327 | 0.1874 | 0.3990 |
| Mapped real-data transfer | `distilbert-base-uncased` | 0.4156 | 0.3515 | 0.5778 | 0.7976 | 0.2182 | 0.3888 |
| Mapped real-data transfer | `tfidf_two_step` | 0.3740 | 0.2303 | 0.5403 | 0.7992 | 0.1162 | 0.7019 |

## A.11 Generator Realism-Validation Protocol

To assess generator quality without redefining the paper's main task, we built a small-scale real-versus-synthetic discrimination protocol. The reference pool is a public OMSCS review source from which 32 reviews were prepared across four computer science courses. This source is useful for a first realism stress test because it provides naturally written educational reviews with concrete course experiences, but it is not representative enough to serve as the only real benchmark. It is graduate-level, computer-science-heavy, English-only, and drawn from a single public review website. Accordingly, OMSCS is treated as a constrained validation source rather than a final external evaluation set, and these reviews are never mixed into the ABSA train, validation, or test splits.

The realism-validation procedure is iterative. In this protocol, one cycle means a complete pass over 60 judge questions: 30 real reviews and 30 synthetic reviews, each judged independently as a binary REAL-versus-SYNTHETIC decision with confidence, cue tags, and justification. Synthetic reviews are generated from sampled aspect sentiments together with a sampled subset of nuance attributes rather than the full schema, so the prompt remains controllable without looking templated. Each cycle records judge labels, confidence, entropy, a chance-confusion percentage, and exact-versus-chance significance statistics. When

the judge correctly identifies synthetic cues, those explanations are fed into an editor step that rewrites the instruction before the next cycle, so each cycle is evaluated under one fixed prompt specification rather than an ad hoc prompt mixture. These checks are positioned as generator validation steps that support the main synthetic-data ABSA study, not as a substitute for the paper's core training-and-evaluation experiments. They are diagnostic, single-source, and single-judge evidence rather than a broad realism claim.

$$\hat{y}_{c,i} \in \{\text{real}, \text{synthetic}\}, \qquad q_{c,i} \in [0, 1]$$

$$\text{Acc}_c = \frac{1}{M} \sum_{i=1}^{M} \mathbf{1}[\hat{y}_{c,i} = y_i], \qquad M = 60$$

$$\text{Confusion}_c = 100 \left( 1 - \frac{|\text{Acc}_c - 0.5|}{0.5} \right)$$

$$H_c = -\frac{1}{M} \sum_{i=1}^{M} \left( q_{c,i} \log q_{c,i} + (1 - q_{c,i}) \log(1 - q_{c,i}) \right)$$

$$I_{c+1} = E\left( I_c, \{(\hat{y}_{c,i}, q_{c,i}, r_{c,i}) : \hat{y}_{c,i} = \text{synthetic} \wedge y_i = \text{synthetic}\} \right)$$

Here $y_i$ is the hidden ground-truth source label, $\hat{y}_{c,i}$ is the judge decision for item $i$ in cycle $c$, $q_{c,i}$ is the judge confidence, $r_{c,i}$ denotes the textual cue explanation, $\text{Acc}_c$ is the judge accuracy, $\text{Confusion}_c$ is the paper-facing chance-confusion statistic, $H_c$ is mean decision entropy, and $E$ is the editor function that produces the next stable realism instruction only from cues that correctly exposed synthetic items.

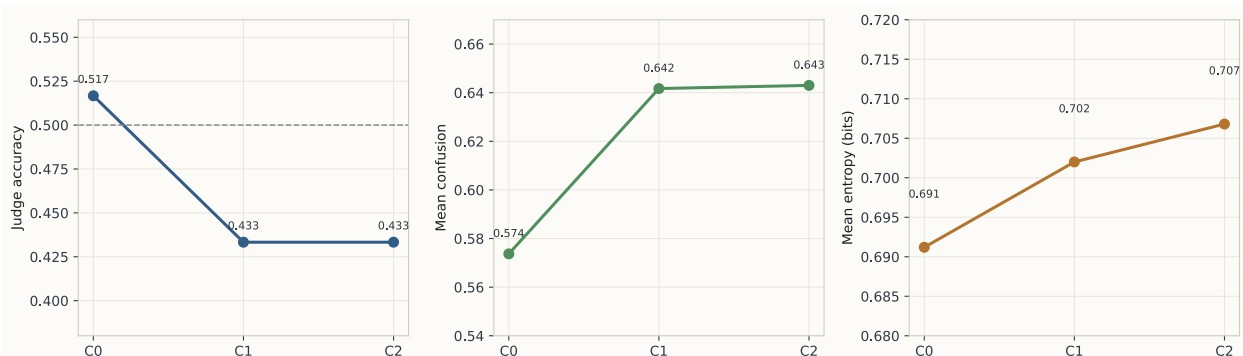

Figure A5. Realism-validation outcomes across three judge-editor cycles. The three panels summarize judge accuracy, chance-confusion, and decision entropy over complete 60-question cycles. The figure documents how judge uncertainty changes across cycles rather than claiming statistical indistinguishability from real reviews.

## A.12 Complementary Generator Diagnostics

- A real-versus-synthetic validation harness with cycle-by-cycle prompt-debug logging.
- Compact model-assisted audits of aspect support and sentiment consistency on both stable-prompt samples and the full dataset.

We also ran compact model-assisted auxiliary audits with `gpt-5.2`. On 30 synthetic reviews generated directly from the corpus-generation prompt, all declared aspects were judged to be textually supported (1.0000 support rates at both the aspect and review level), while exact sentiment agreement was lower (0.7377 at the aspect level and 0.5333 at the full-review level). This prompt-level result should be read together with the harsher full-dataset audit reported in Section 5.6: the combination suggests that target aspect presence can be preserved under the generation prompt, but large-scale generation still introduces substantial label-faithfulness drift, especially for sentiment polarity. Together these diagnostics justify treating realism and faithfulness checks as generator audits rather than as evidence that the 10K corpus is a synthetic gold standard.

### A.13 Three-Cycle Realism-Validation Results

The realism study remains diagnostic, and it is reported here as a complete three-cycle procedure rather than as a single pass. Each cycle used 60 independent judge questions: 30 real OMSCS reviews and 30 synthetic reviews generated from the richer attribute prompt, with binary REAL-versus-SYNTHETIC decisions, confidence scores, cue tags, and textual justification. Across cycles 0, 1, and 2, judge item accuracy was 0.5167, 0.4333, and 0.4333; mean confusion was 0.5737, 0.6417, and 0.6430; mean entropy was 0.6912, 0.7020, and 0.7068 bits; and the number of correctly detected synthetic reviews fell from 5 to 1 to 0. These diagnostics indicate that the stabilized prompt reduced several synthetic cues, even though the accuracy-based chance-confusion statistic was not monotonic across cycles. Figure A5 plots the per-cycle accuracy, confusion, and entropy, and Table A12 summarizes the three cycles.

The final cycle does *not* support an equivalence claim. In cycle 2, the judge accuracy remained 0.4333, the exact binomial test against chance yielded `p = 0.366294`, and the Wilson 95% interval was `[0.3157, 0.5590]`, which is too wide for the predefined $\pm 0.10$ equivalence margin. The value of the procedure is therefore diagnostic rather than confirmatory. Judge explanations repeatedly highlighted generic specificity, overbalanced structure, overpolished prose, stacked motifs, and occasional persona inconsistencies; those cues were then used by the editor step to stabilize the next-cycle prompt. The realism analysis therefore contributes a cycle-level improvement curve, cue summaries, prompt states, and side-by-side appendix examples rather than only a final binary verdict.

Table A12. Realism-study summary across the three complete judge-editor cycles, including accuracy, confusion, entropy, and whether prompt rewriting was triggered.

| Prompt state | Cycle | Judge accuracy | Mean confusion | Mean entropy (bits) | Binomial p-value | Editor triggered |
|---|---|---|---|---|---|---|
| `rich_attributes_ baseline` | 0 | 0.5167 | 0.5737 | 0.6912 | 0.8974 | yes |
| `reduce_synthetic_ signatures` | 1 | 0.4333 | 0.6417 | 0.7020 | 0.3663 | yes |
| `messier_realism` | 2 | 0.4333 | 0.6430 | 0.7068 | 0.3663 | no |

To make the realism analysis auditable, the appendix includes side-by-side real and synthetic review examples together with the synthetic review's sampled aspects and nuance attributes. The paper therefore reports not only the final cycle summary, but also the prompt instructions and qualitative evidence used to justify revision between cycles; the paired examples appear in Appendix A.2.

### A.14 Full-Corpus Output Control

The full dataset is large enough to support a realistic internal benchmark, and it also reveals how output control behaves at scale. All 10,000 rows returned usable review text and no duplicates were detected. At the same time, 841 rows were marked incomplete because they reached the output-token cap, and overall length-band adherence settled at 0.6819 over the full corpus. The resulting dataset is therefore broad and usable while still leaving room for stronger long-form control in later releases.

This behavior does not invalidate the benchmark, because every row still contains the target labels and usable text, and the train-validation-test pipeline runs cleanly on the assembled corpus. It does, however, matter for interpretation: stronger long-form length control would further improve the controllability of future releases. Figure A7 makes this visible directly by combining length, aspect-count, course, and style evidence in one place.

### A.15 Corpus Profile and Representative Examples

Figure A6 and Figure A7 establish the corpus-level structure before any model results are introduced. Figure A6 shows how label support is distributed across aspects and sentiment polarities, which makes it easier to distinguish heavily represented pedagogical dimensions from thinner parts of the schema. Figure A7 complements that view with review length, aspect-count mixture, course coverage, and style variation. Table

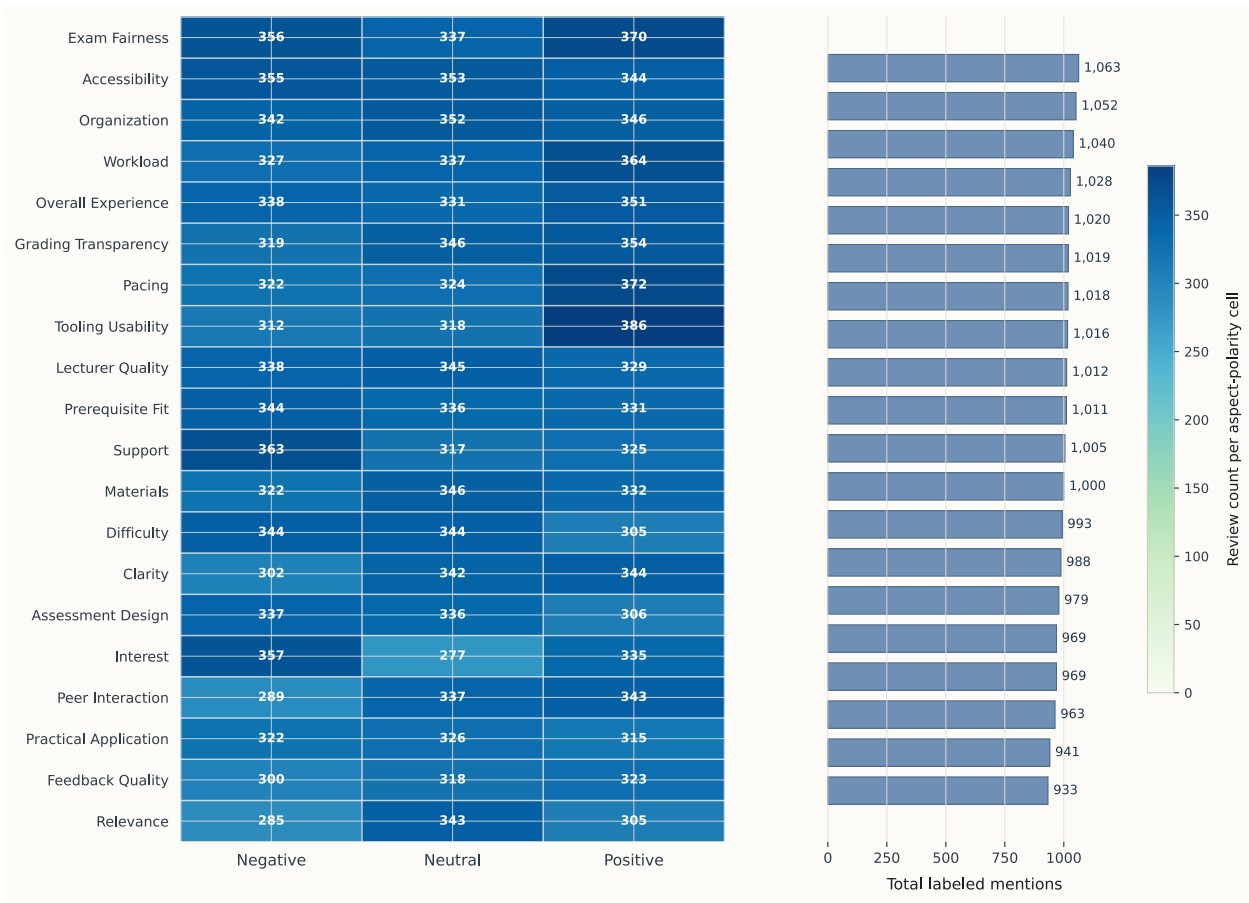

Figure A6. Aspect-level label support across the 10K / 20-aspect corpus. The heatmap shows polarity composition for each aspect, ordered by total support, and the companion bar panel shows the total number of labeled mentions per aspect.

A13 then gives a qualitative sense of how these distributions appear in actual review text. These examples are included to help the reader interpret the benchmark qualitatively, not as a substitute for aggregate model evidence.

Table A13. Representative synthetic review snippets showing how aspect labels appear under different writing styles and pedagogical situations.

| Example type | Style | Aspect labels | Excerpt |
|---|---|---|---|
| Tradeoff-heavy | Neutral everyday prose | `difficulty=positive, materials=positive` | "I took Computer Networks mainly to fit my schedule ... the material felt accessible at times ... Dr. Chen's delivery varies week to week ... the LMS was smooth, but feedback came too late to be useful." |
| Compressed complaint | Analytic but Simple | `interest=negative` | "I stuck with Cyber Security Basics only when I needed it ... the lead-in theory never clicked for me ... the few but heavy deadlines left me frustrated but still fair about the effort." |
| Pedagogical frustration | Analytic but Simple | `assessment_ design=negative, relevance=negative` | "Data Structures felt like a required hurdle more than a doorway to real understanding ... the assessments were exhausting and overbearing ... even when I passed a task, I was not convinced it reflected useful learning." |

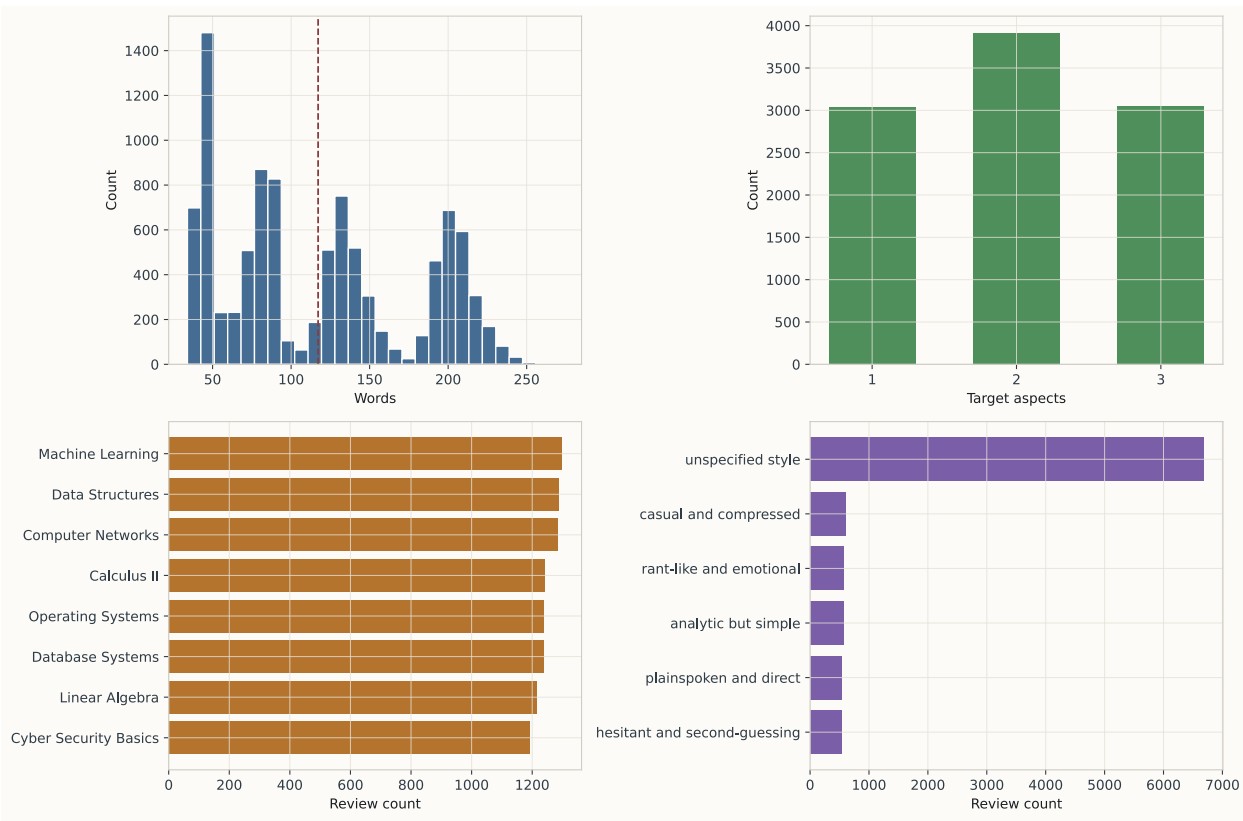

Figure A7. Corpus profile and diversity cues. The panels visualize the distributional shape of review length, aspect-count mixture, course coverage, and observed style cues, while Table 4 provides the corresponding exact summary values.

### A.16 Per-Aspect Behavior of the Best Model

The per-aspect profile of the best untuned model, BERT, is uneven in a way that matches the pedagogical complexity of the label space. Stronger aspect categories concentrate in the assessment-and-management and learning-demand blocks, especially `workload`, `grading_transparency`, `exam_fairness`, `pacing`, and `tooling_usability`. Harder categories cluster in the instructional-quality, learning-environment, and engagement blocks, including `peer_interaction`, `support`, `interest`, `feedback_quality`, and `clarity`, which likely require more implicit reasoning and weaker lexical cues. This pattern shows that the benchmark is not uniformly easy even when labels are known by construction. Figure A8 plots the per-aspect detection and sentiment profile, and Table A14 lists the strongest and weakest aspect categories.

Table A14. Strongest and weakest aspect categories for the best local model, showing where the benchmark is comparatively explicit or comparatively latent.

| Model | Group | Aspect | F1 | Sentiment MSE | Precision | Recall |
|---|---|---|---|---|---|---|
| bert-base-uncased | top-5 | workload | 0.5864 | 0.3727 | 0.6437 | 0.5385 |
| bert-base-uncased | top-5 | grading_transparency | 0.5513 | 0.5838 | 0.8958 | 0.3981 |
| bert-base-uncased | top-5 | exam_fairness | 0.5316 | 0.3735 | 0.7500 | 0.4118 |
| bert-base-uncased | top-5 | pacing | 0.5093 | 0.3199 | 0.5000 | 0.5189 |
| bert-base-uncased | top-5 | tooling_usability | 0.4873 | 0.4892 | 0.5333 | 0.4486 |
| bert-base-uncased | bottom-5 | clarity | 0.2006 | 0.5926 | 0.1166 | 0.7172 |
| bert-base-uncased | bottom-5 | feedback_quality | 0.1961 | 0.5917 | 0.1117 | 0.8049 |
| bert-base-uncased | bottom-5 | interest | 0.1814 | 0.5322 | 0.1114 | 0.4889 |
| bert-base-uncased | bottom-5 | support | 0.1805 | 0.6977 | 0.1049 | 0.6458 |
| bert-base-uncased | bottom-5 | peer_interaction | 0.1385 | 0.8416 | 0.3913 | 0.0841 |

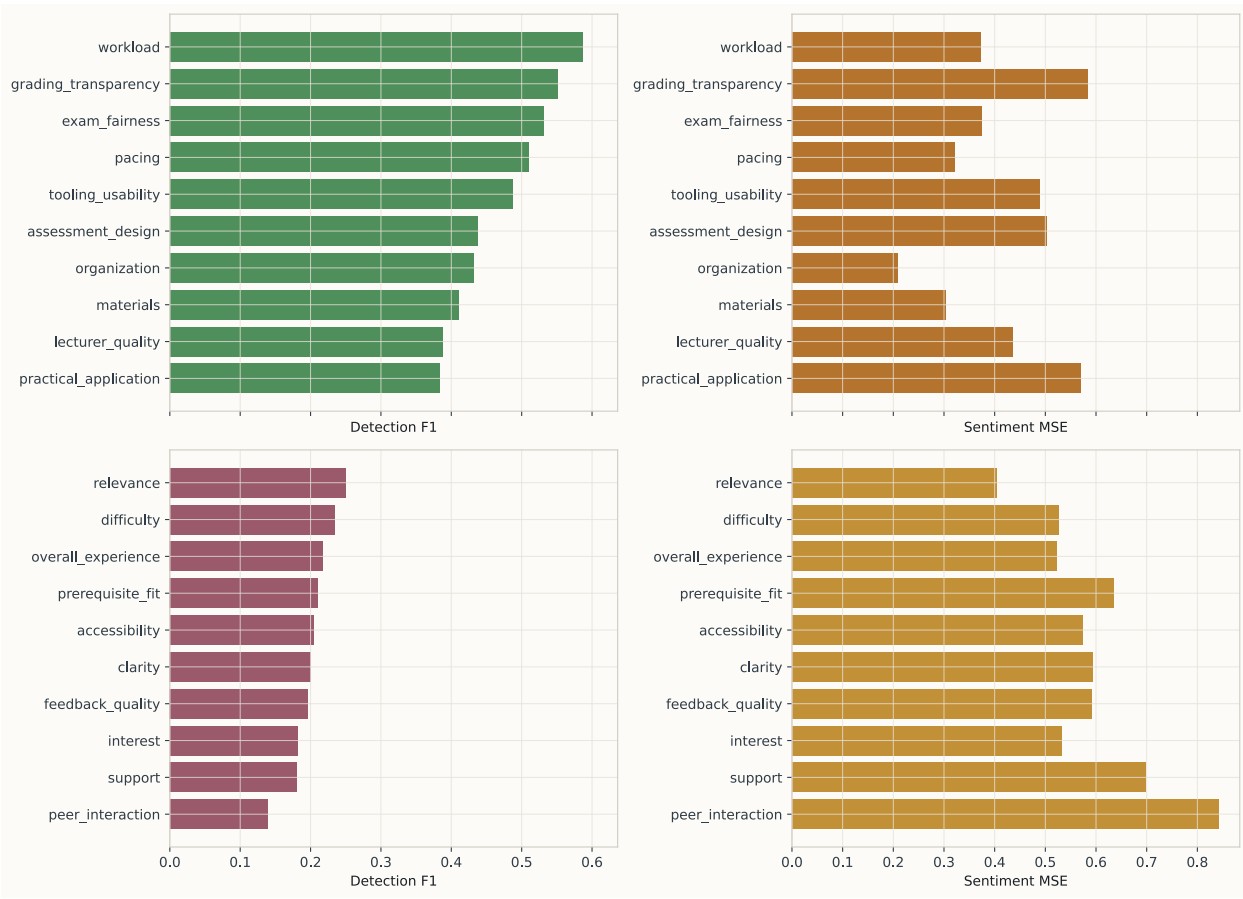

Figure A8. Per-aspect detection F1 and sentiment MSE for the best untuned local model, BERT, on the held-out test split. The figure separates the strongest and weakest aspect groups, making it easier to see where the benchmark remains lexically explicit versus pedagogically latent.

### A.17  Prior Real-Data Educational Sentiment Results in Context

Table A15 is included to prevent overinterpretation of the transfer result. Our mapped-overlap score should not be read as "worse than Herath" or "better than Welch" in any simple leaderboard sense, because the task definitions differ substantially. Welch and Mihalcea study targeted sentiment toward extracted course or instructor entities, while Herath et al. evaluate aspect-level sentiment under their own annotation structure. The most defensible reading is narrower: the reported result shows that models trained only on the synthetic corpus recover some real educational signal on one mapped real benchmark, but the paper does not yet claim parity with prior real-data educational sentiment systems under their native tasks.

Table A15. Prior real-data educational sentiment results included for context; the final column states how directly each study compares to the present benchmark.

| Study | Setting | Reported metric | Task | Comparability to this study |
|---|---|---|---|---|
| Welch and Mihalcea [7] | Real student comments with automatically extracted entities | F1 = 0.586 | Targeted sentiment toward courses and instructors | Low; entity-targeted sentiment is narrower than review-level 20-aspect ABSA |
| Welch and Mihalcea [7] | Real student comments with ground-truth entities | F1 = 0.695 | Targeted sentiment with gold entities | Low; useful upper-bound reference, but not a directly matched benchmark |
| Herath et al. [11] | Real student feedback baseline | F1 = 0.750 | Aspect-level sentiment analysis | Medium; closest real educational sentiment benchmark, but different label unit and task structure |
| Herath et al. [11] | Real student feedback best tree-based ALSC | F1 = 0.710 | Aspect-level sentiment classification with FT-RoBERTa induced RGAT | Medium; stronger educational baseline under their own annotation protocol |
| This study | Mapped Herath overlap after synthetic training | micro-F1 = 0.4593 | Synthetic-to-real transfer on a 9-aspect overlap | Medium; the closest comparison here because it uses real educational reviews, but it still depends on conservative aspect mapping |

