# OpenReview forum: "A Controlled Synthetic Benchmark for Educational Aspect-Based Sentiment Analysis"
_TMLR — Under review for TMLR_

### Review · Reviewer_dWED · 2026-07-01

**Summary Of Contributions:**

This paper presents a synthetic dataset and methodology for educational Aspect-Based Sentiment Analysis (ABSA). The key contributions are:
1.	A 10,000-review synthetic corpus with a 20-aspect pedagogical schema, generated through a controlled protocol that separates supervision targets from nuance attributes, enabling varied course contexts and writing styles.
2.	A faithfulness-aware data quality pipeline using cost-matched LLM audits to score per-label faithfulness, validated against human labels (Cohen's kappa 0.56) and used as a training-data filter.
3.	A signal-validity battery with label-permutation, learning-curve, and clean-label-ceiling controls demonstrating that the benchmark carries learnable signal beyond label frequencies.
4.	Real-data transfer evidence showing synthetic-only training recovers ~60% of a real-trained model's performance, and synthetic pre-training followed by real fine-tuning exceeds real-only training.
Key Strengths
1.	Rigorous experimental design with strict data splits, multi-seed replication, and comprehensive control experiments that definitively establish learnable signal beyond memorized priors.
2.	Novel faithfulness-aware filtering pipeline validated three ways—behavioral negative control, cross-provider judge agreement, and human-label validation —demonstrating that retaining only the top 50% of the corpus reduces sentiment-prediction error while matching full-corpus performance at half the training cost.
3.	Comprehensive evaluation across diverse architectures spanning classical TF-IDF, four transformer encoders, joint prediction variants, and four GPT-based inference methods, all under identical strict conditions, providing a meaningful benchmark for future work.
4.	Pedagogically meaningful 20-aspect schema organized into five actionable teaching dimensions that map directly to institutional intervention points.
5.	Strong transfer learning evidence showing synthetic pre-training followed by real fine-tuning exceeds real-only training, demonstrating genuine learned representations rather than just surface patterns.
Key Weaknesses
1.	Generator-auditor circularity concerns as the same provider family (OpenAI) generates the data and performs the faithfulness audit, raising residual questions about whether the audit detects generator artifacts rather than true textual faithfulness despite cross-provider and human validation.
2.	Moderate absolute performance (best micro-F1 of 0.276, sentiment MSE ~0.50) that, while reflecting task difficulty, leaves unclear what "good enough" thresholds are for practical educational dashboards.
3.	Insufficient analysis of the 841 incomplete rows (8.4% of corpus) that hit token caps, with no examination of whether these are systematically biased across aspects or sentiment polarities.
4.	Limited qualitative error analysis showing common failure modes, making it difficult for practitioners to diagnose whether errors are systematic rather than random.
5.	Narrow generalizability evidence limited to English STEM graduate contexts, with no validation across K-12, non-STEM domains, other languages, or different institutional settings.

**Audience:**

Yes

**Audience Explanation:**

This paper would interest the TMLR audience for several reasons:
1.	Methodological contribution: The faithfulness-aware filtering pipeline is generalizable beyond educational ABSA, relevant to researchers working with LLM-synthesized supervision.
2.	Domain relevance: Educational feedback analysis is an active area where labeled data scarcity is a known bottleneck—the resource fills a genuine gap.
3.	Transfer learning insights: The finding that synthetic pre-training can exceed real-only training is practically valuable.
4.	Reproducibility: The paper provides a well-documented, reproducible benchmark with artifacts and code, which is highly valuable for the community.

**Broader Impact Concerns:**

The paper includes an ethics statement that appropriately addresses key concerns:
•	No identifiable student data: The synthetic corpus contains no real student information.
•	Proper use of public data: The Herath corpus is used under its MIT license; OMSCS reviews are public.
•	Explicit deployment guidance: The paper clearly states the resource is appropriate for low-stakes, aggregate feedback and inappropriate for high-stakes personnel decisions without human review.

**Claims And Evidence:**

Yes

**Claims Explanation:**

The paper provides extensive, well-structured evidence:
•	Comprehensive benchmark evaluation across multiple architectures with clear metrics and statistical reporting
•	Rigorous experimental controls including label permutation, training size scaling, and faithfulness-based filtering
•	External validation on two real corpora with conservative aspect mappings
•	Human validation of the faithfulness audit and cross-provider LLM agreement
•	Detailed ablation studies showing the filtering recipe works across architectures and targets
The evidence is convincing, though the absolute performance scores are modest—which the authors acknowledge as reflecting task difficulty rather than model weakness.

**Requested Changes:**

1.	Clarify the relationship between generator and auditor: The paper uses GPT-5-nano as generator and GPT-5.2/gpt-4.1-mini as auditor, both from the same provider family. While cross-provider and human validation help, the paper should more explicitly discuss potential circularity—is the audit simply detecting patterns from the same model family's latent space rather than true textual faithfulness? A dedicated discussion of this limitation is needed.
2.	Address the 841 incomplete rows: The paper notes 841 rows hit the output-token cap but doesn't fully discuss whether this creates systematic bias (e.g., shorter reviews having different aspect distributions or sentiment profiles than longer ones). Add an analysis of whether these incomplete rows are distributed across aspect categories and whether excluding them changes benchmark results.
3.	A broader discussion: The article mentions LLM prompts, many novel articles and methods have emerged in this field at present, including Seeing Sarcasm Through Different Eyes: Analyzing Multimodal Sarcasm Perception in Large Vision-Language Models and Exploiting the Relationship within the Unlabelled Samples by Set Matching for Generalized Category Discovery.
4.	Strengthen the discussion of transfer limits: The real-data transfer uses only 9 of 20 aspects, and performance is ~60% of real-trained models. The paper should more prominently state what practitioners should not conclude—specifically, that the full 20-aspect schema lacks real validation and that high-stakes decisions require human-in-the-loop review.
5.	Add a clearer practitioner roadmap: Given the moderate faithfulness rates (0.42 sentiment-match), provide concrete, actionable guidance for institutions that might adopt this resource: minimum fine-tuning data size, expected performance degradation on their own data, and monitoring requirements.

---

### Review · Reviewer_h7LN · 2026-07-03

**Summary Of Contributions:**

In this study, the authors address the use of large language models (LLMs) for generating synthetic data in the context of aspect-based sentiment analysis (ABSA) within educational evaluation.
- A synthetic ABSA corpus comprising 10,000 course reviews has been compiled and released.
- To mitigate the label inconsistency commonly observed in LLM-generated data, a cost-matched LLM auditing mechanism is introduced. This quantifies a fidelity score for each label, which is subsequently used to filter the training data.
- A rigorous evaluation matrix for assessing the "signal validity" of synthetic data has been devised. This is intended to demonstrate that the synthetic corpus contains genuinely learnable semantic signals, rather than merely reflecting label frequency distributions that a model might memorise.

**Additional Comments:**

See the above comments and concerns.

**Audience:**

Yes

**Audience Explanation:**

The sentiment evaluation and ABSA research communities are likely to take an interest in this paper.

**Broader Impact Concerns:**

- The real OMSCS reviews used for authenticity validation number a mere 32, all drawn from computer science master's courses; the external transfer dataset from Herath comprises only 2,829 reviews. The experimental conclusions are thus applicable to an extremely narrow scope.

**Claims And Evidence:**

No

**Claims Explanation:**

- The manuscript claims to have constructed a controlled synthetic corpus; however, 841 of the 10,000 generated samples hit the token limit and were truncated, yielding a full-dataset length constraint match rate of merely 0.6819 (Appendix A14). A substantial proportion of samples therefore fall outside the pre-specified length bounds.

- The utility of low-fidelity samples is open to question. The bottom 25% instances—the worst-scoring quartile—appear to markedly inflate sentiment prediction error (as evidenced by the discrepancies in Table 12). Such data would seem to hold little, if any, training value.

- Table 10 reports the audit results for a random sample of 250 entries (501 annotated aspects) drawn from the 10K corpus: aspect sentiment-match rate = 0.4232 ≈ 0.42. In plain terms, for every 100 human-assigned aspect–sentiment pairs, only 42 find corresponding emotional expression in the generated text; over half the assigned sentiments conflict with or lack support in the actual content.

- The LLM audit achieves only moderate agreement with human annotations: Cohen's kappa = 0.56 against human labels (moderate agreement by Landis–Koch criteria), so the audit scores cannot be taken as a fully reliable proxy for human judgement.

**Requested Changes:**

- The assessment of synthetic text "realism" relies chiefly on LLMs acting as their own judges. Whilst this approach has been refined through prompt engineering, it fails to establish statistically that these synthetic reviews are indistinguishable from genuine student feedback. Moreover, although the label fidelity audit achieves a Kappa of 0.56 against human annotations, the full set of dimensional labels remains without comprehensive human re-annotation verification.
- In baseline comparisons with other large models, testing is confined to a single provider's GPT family and a single structured prompting methodology. It does not extend to the broader landscape of mainstream closed-source or open-source alternatives currently available.
- The formatting of Figure 1, Table 5, Figure A2 and Figure A3 could benefit from further refinement.

---

### Review · Reviewer_nfat · 2026-07-19

**Summary Of Contributions:**

This paper presents a 10,000-review synthetic benchmark for educational aspect-based sentiment analysis with 20 pedagogical aspects. It separates target labels from contextual attributes during generation, audits label faithfulness using LLM judges, and uses audit scores to filter training data. The paper also includes sanity checks, synthetic-to-real transfer experiments, and synthetic pretraining followed by real-data fine-tuning. Strengths include a relevant low-resource application, careful analysis of synthetic label noise, useful permutation and learning-curve controls, and an actionable filtering idea. The main weakness is that label quality remains low. The strict audit reports only 0.423 aspect-sentiment agreement and 0.212 full-row agreement. Therefore, the corpus is better viewed as noisy synthetic supervision than as a reliable benchmark.

**Audience:**

Yes

**Audience Explanation:**

The paper addresses an important problem in LLM-generated supervision: generated text may appear realistic while failing to support its assigned labels. The audit-filter-validate framework is relevant to synthetic-data research beyond educational ABSA. The transparency about label noise and the transfer experiments are also valuable. A revised version with direct human validation and stronger evaluation controls could be useful to TMLR readers.

**Broader Impact Concerns:**

The paper appropriately discourages high-stakes use, but the broader-impact discussion should more clearly address instructor evaluation, privacy of student comments, bias against non-native or unusual writing styles, and the risk of attaching fictional negative reviews to identifiable courses or instructors. Institutional deployment should require human review, uncertainty reporting, data protection, and an appeal process.

**Claims And Evidence:**

No

**Claims Explanation:**

The experiments show that the corpus contains some learnable signal. Label permutation collapses performance, more training data improves results, and synthetic training transfers partially to real data.

However, the main claims are not fully supported. First, most synthetic rows contain at least one unsupported sentiment label. Second, the auditor is validated mainly using artificially perturbed real labels rather than direct human annotation of synthetic reviews. Third, sentiment MSE is measured only on aspects predicted by each model, so filtering comparisons may be affected by different prediction sets. Fourth, random splits from the same generator and prompt may reward generator-specific patterns. Finally, some results are inconsistent, including the BERT scores in Tables 8 and 9 and aspect-count totals that do not sum to 10,000.

**Requested Changes:**

Human-annotate a representative sample of the actual synthetic corpus and report aspect and sentiment agreement.
Match filtering subsets by aspect, polarity, aspect count, length, and style to isolate the effect of faithfulness.
Resolve inconsistent numbers, especially Tables 8 and 9, the aspect-count totals, and the incomplete abstract.
Shorten repeated discussion and clearly describe the resource as noisy synthetic supervision rather than a gold benchmark.